# Evaluating and Improving Tool-Augmented Computation-Intensive Math Reasoning

**Beichen Zhang**[13*], **Kun Zhou**[23*], **Xilin Wei**[4], **Wayne Xin Zhao**[13†],
**Jing Sha**[5], **Shijin Wang**[56], **Ji-Rong Wen**[123]
[1]Gaoling School of Artificial Intelligence, Renmin University of China.
[2]School of Information, Renmin University of China.
[3]Beijing Key Laboratory of Big Data Management and Analysis Methods.
[4]College of Computer Science, Sichuan University.
[5]iFLYTEK Research, [6]iFLYTEK AI Research (Central China).
{zhangbeichen724,wiselnn570,batmanfly}@gmail.com, francis_kun_zhou@163.com
{jingsha,sjwang3}@iflytek.com,jrwen@ruc.edu.cn

## Abstract

Chain-of-thought prompting (CoT) and tool augmentation have been validated in recent work as effective practices for improving large language models (LLMs) to perform step-by-step reasoning on complex math-related tasks. However, most existing math reasoning datasets may not be able to fully evaluate and analyze the ability of LLMs in manipulating tools and performing reasoning, as they often only require very few invocations of tools or miss annotations for evaluating intermediate reasoning steps, thus supporting only outcome evaluation. To address the issue, we construct **CARP**, a new Chinese dataset consisting of 4,886 computation-intensive algebra problems with formulated annotations on intermediate steps, facilitating the evaluation of the intermediate reasoning process. In CARP, we test four LLMs with CoT prompting, and find that they are all prone to make mistakes at the early steps of the solution, leading to incorrect answers. Based on this finding, we propose a new approach that can facilitate the deliberation on reasoning steps with tool interfaces, namely **DELI**. In DELI, we first initialize a step-by-step solution based on retrieved exemplars, then iterate two deliberation procedures that check and refine the intermediate steps of the generated solution, from both tool manipulation and natural language reasoning perspectives, until solutions converge or the maximum iteration is achieved. Experimental results on CARP and six other datasets show that the proposed DELI mostly outperforms competitive baselines, and can further boost the performance of existing CoT methods. Our data and code are available at `https://github.com/RUCAIBox/CARP`.

## 1 Introduction

Recently, large language models (LLMs) (*e.g.,* GPT-3 and ChatGPT) have shown remarkable zero-shot and few-shot performance on various tasks [1–6], including language generation and question answering. As LLMs have been pre-trained on a vast amount of text data that covers a broad spectrum of world knowledge, existing work also shows that LLMs can solve complex tasks, *e.g.,* math reasoning [7–12] and college entrance exam [4, 13].

To evaluate the capacity of LLMs for solving complex tasks, math reasoning datasets have been widely used as testbeds, *e.g.,* GSM8K [14] and MATH [15], where the math problems can not

---

*Equal contributions.
†Corresponding author.

37th Conference on Neural Information Processing Systems (NeurIPS 2023) Track on Datasets and Benchmarks.

be directly answered but require multi-step reasoning. To elicit LLMs for step-by-step reasoning, chain-of-thought (CoT) [7, 16–20] has become the de facto prompting strategy, where LLMs can be guided to generate a solution consisting of a series of intermediate steps to reach the answer. However, previous work also reveals that LLMs are prone to make mistakes at intermediate steps, especially in numerical computation [10, 21–24], and yet a minor mistake would lead to an incorrect final answer. To alleviate it, a line of work [25–40] employs external tools to make up for the weakness of LLMs, and can greatly improve accuracy of answers on math reasoning tasks. With the rapidly evolving LLMs and tool-augmented methods, it is necessary to adopt a suitable math reasoning dataset for evaluating them systematically and differentially. Whereas, problems in most existing math reasoning datasets may only require the one-off utilization of tools [26, 27, 29, 30], which are not adequate to fully measure the ability of tool manipulation in existing methods. Moreover, even though incorrect answers often stem from errors in intermediate reasoning steps, most existing datasets are not equipped to test this, due to a lack of formal annotations of the intermediate steps in the solution text. The two issues limit existing math reasoning datasets to systemically evaluate and analyze LLMs and tool-augmented methods.

To address them, we construct a new Chinese dataset that consists of 4,886 **C**omputation-intensive **A**lgeb**R**a **P**roblems associated with formulated annotations of all the intermediate steps, namely **CARP**. In CARP, problems require deriving multiple intermediate math expressions based on math knowledge, and solving them based on arithmetic knowledge, which makes it a complex and difficult dataset to evaluate the computation-intensive math reasoning ability. In addition, the formulated annotations also enable researchers to test the accuracy of intermediate reasoning steps for analyzing the errors of LLMs. As shown in Table 3, four popular LLMs with CoT prompting can not solve over half of the problems in our CARP, indicating the difficulty of CARP. Furthermore, we also find that all LLMs are more likely to make mistakes in the first step (over 69%), leading to totally wrong solutions and answers. It reveals that LLMs mostly fail in performing early reasoning steps, and can not correct the errors in the latter steps. Based on CARP, we also devise a variety of fine-grained interfaces based on available tools, to provide practical functionalities for handling complicated calculations. These interfaces can also be applied to other math reasoning datasets to improve the tool manipulation capacity of LLMs.

Considering that LLMs can not fix errors in early steps by themselves in the generation, we propose a new approach that can **deli**berate [3] the reasoning steps of LLMs with tool interfaces, namely **DELI**. DELI can iteratively refine the generated result from both tool manipulation and natural language reasoning, which makes it able to correct the mistakes of existing solutions from different perspectives and avoid error accumulation. Concretely, we initialize a step-by-step solution for the given question based on retrieved relevant exemplars, then iterate two deliberation procedures that check and refine the generated solution from the perspectives of tool manipulation and natural language reasoning, until reaching the stop condition, *e.g.,* solution has converged or iterations reach the maximum number. Such a way is similar to the solution checking process of humans, and can elicit LLMs to deliberate and correct the possible errors in intermediate steps of the solution. We evaluate DELI and existing prompting methods on CARP and six other computation-intensive datasets. Experimental results show that DELI mostly outperforms competitive baselines (*e.g.,* 9.35% accuracy improvement over the best baseline on CARP), and can further boost the performance of existing CoT prompting methods.

To summarize, our major contributions are:

• We construct a new dataset named CARP with formulated annotation of intermediate reasoning steps for systematically evaluating LLMs in solving computation-intensive math problems, and devise interfaces with practical functionalities to help LLMs.

• We propose DELI, a new approach that can deliberate and correct the reasoning steps of LLMs with tool interfaces.

• We conduct extensive experiments to show the superiority of our DELI over existing prompting methods on 7 computation-intensive math reasoning datasets.

---

[3]In this work, we use the term "Deliberation" to reflect that the LLM can self-check and refine its generated solution iteratively until the result converges. This iterative refinement bears resemblance to the human cognitive process of meticulous contemplation and correction of potential errors in problem solving.

## 2 Related Work

**Math Problem Solving Datasets.** A line of work constructs math word problem datasets with annotations of several math equations [41–52, 35, 53]. With the growth of the language models' reasoning ability, researchers propose more challenging math word problem datasets annotated with natural language solutions. GSM8k [14] is a diverse math problem dataset with multi-step reasoning annotation. However, the above math word problem datasets do not involve complex math domain knowledge, and the computation types are limited. MATH [15] is a challenging dataset at high school competition difficulty, covering various math domains. Besides, math subsets of AGIEval [13] collect problems requiring complex reasoning and computations from college entrance exams for humans. Our proposed CARP dataset mainly differs from these computation-intensive datasets in two aspects: (1) We provide formulated annotations of intermediate reasoning steps, thus supporting both process and outcome evaluation; (2) We provide fine-grained interfaces for evaluating the tool manipulation ability of LLMs in complex reasoning.

**Tool-Augmented Language Models.** To expand the capability boundaries of language models, a line of work employs external tools to aid them [25–40]. Existing work typically employs LLMs to determine tool arrangement or generate programs, then executes them in a single pass to obtain the results for solving math word problems [26, 27, 29, 30, 34]. Nevertheless, computation-intensive math problems encompass a range of challenges, including math domain knowledge, math expression understanding, and complex computations. The *plan-and-execute* approaches, which lack a procedure of reasoning based on intermediate results, may not be sufficient for handling such problems. Re-Act [32] is a tool-augmented approach that interleaves reasoning and tool manipulation in the solving process. We extend ReAct to the math domain by incorporating our fine-grained tool interfaces as a component of our iterative deliberation approach.

**Iterative Refinement Methods.** Large language models have demonstrated remarkable performance in math-related tasks [7, 17, 16, 54–60]. Chain-of-thought (CoT) prompting is notably effective for complex reasoning in LLMs [7, 16]. However, LLMs are prone to make minor mistakes in multi-step reasoning [61]. To this end, existing studies explore iteratively refining reasoning steps with LLMs [62–71]. A line of work leverages self-generated feedback to improve math problem solutions [63–65]. However, recent studies show that LLMs may struggle with identifying problems in the text or providing faithful explanations [61, 72, 73]. Therefore, minor errors, such as inaccurate computation, pose a challenge for LLMs to check and correct solutions when solving computation-intensive problems. Another line of work explores iterative methods for solving computation-intensive math problems. Learning to Program [74] aims to iteratively learn natural language problems from training sets to guide the solving process. Progressive-Hint Prompting [69] is an iterative method that leverages prior answers as hints to guide LLMs in generating better CoT solutions.

## 3 CARP Dataset

Computation-intensive math reasoning task aims to solve complex math problems that require performing multi-step arithmetical computation and reasoning based on mathematical knowledge. Typically, to solve a computation-intensive math problem, humans or models need to iteratively derive the math expression and then compute the result (via calculators or other tools), until obtaining the final answer. In this way, the accuracy of intermediate reasoning and computation steps is crucial, where a subtle error would lead to completely wrong answers. In this paper, we construct a new dataset CARP (Computation-intensive AlgebRa Problems) that provides the formulated annotations of all the intermediate steps for the computation-intensive middle school math problems. Based on the annotations, we also design a set of interfaces with fine-grained computation functions, to help LLMs manipulate commonly-used tools for solving these problems.

### 3.1 Dataset Construction

Although there are a number of computation-intensive math problems in available datasets, their solutions are generally not well-formulated natural language and may omit intermediate steps [41–52, 35, 13]. To construct a well-formulated dataset, we first collect real-world computation-intensive math problems, and then invite crowdworkers to extract and annotate their expression flow graph.

Table 1: An example from CARP, which is translated into English. Errors are marked in red.

| | |
|---|---|
| **Problem** | The solution to the equation $ax + 2bx = 3$ is $x = 1$ , then the solution to the equation $a(y-1) + 2b(y-1) = 3$ is ? |
| **Solution** | From the question we have : $a + 2b = 3$ , $a(y-1) + 2b(y-1) = 3$ . Rectifying gives $(a + 2b)y - (a + 2b) = 3$ , *i.e.,* $3y - 3 = 3$ , therefore $y = 2$ . |
| **ChatGPT** | Substituting $x = 1$ into $ax + 2bx = 3$ gives $a + 2b = 3$ , and substituting $y - 1$ gives $a(y-1) + 2b(y-1) = 3$, which simplifies to $ay + by = 3$ . $\cdots\cdots$ The answer is $y = \frac{3}{3-a} - 1$ |
| **Error Type** | Reasoning error. |

Table 2: Statistics for CARP dataset.

| Statistic | Number |
|---|---|
| # of training samples | 3,410 |
| # of development samples | 500 |
| # of testing samples | 976 |
| # of nodes (Avg./Max) | 6.0/18 |
| # of edges (Avg./Max) | 5.7/25 |
| # of expression nodes (Avg./Max) | 4.7/15 |
| Problem length (Avg./Max) | 52.1/257 |
| Solution length (Avg./Max) | 71.3/278 |

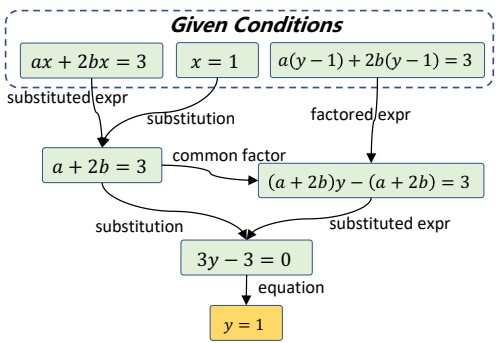

Figure 1: An EFG annotation example for CARP.

**Data Collection.** We collect the math problems and their step-by-step solutions from a Chinese education website Zhixue[4], which contains vast problems to provide education assistance for students. We mainly crawl middle school math problems, since they are of moderate difficulty and require basic arithmetic computations (*e.g.,* quadratic equation) and mathematical knowledge (*e.g.,* Veda's theorem), making them a good testbed for computation-intensive math reasoning. We first crawl about 1,000,000 problems with solutions. Then, to obtain computation-intensive problems, we design hand-crafted rules based on SymPy to roughly extract and count the computation steps in solutions, and only select the ones with both over one computation step and over two reasoning steps. Finally, we invite math teachers to select about 10,000 high-quality examples for annotation.

**Expression Flow Graph Annotation.** In a math problem, the natural language solution can be generally formulated as a directed acyclic graph (DAG), where the nodes and edges refer to the intermediate results and derivation steps, respectively [75]. For computation-intensive problems, we consider a special DAG format that adopts intermediate math expressions as nodes. We name it *expression flow graphs* (EFG), as it can explicitly show how to derive new math expressions based on existing ones in the step-by-step reasoning process. In this way, a solution can be formulated as: starting from initial condition nodes within the problem, we continue deriving new nodes (*i.e.,* intermediate math expressions) from existing nodes, until reaching the final expression that can obtain the answer, where the computation results of intermediate expressions can be utilized for evaluation. Whereas, a math problem may involve special initial conditions that are hard to be converted into readable expressions, *e.g.,* Equations have rational solutions. Thus, we add a special type of node to store these conditions in natural language, while guaranteeing that all the derived new nodes are math expressions. As an example, the EFG annotation of the Problem in Table 1 is shown in Figure 1.

Based on the above definition, we invite five middle school math teachers to crowdsource the annotations of formulated EFGs for collected problems. The annotation process is similar to the information extraction process [76], where we first extract the nodes and then link them to compose the graph. Concretely, we first rely on hand-crafted rules to automatically extract the math expressions and text conditions from the solution texts as node candidates. Then, we ask math teachers to link the related node candidates and annotate their corresponding relations. To reduce the difficulty, we utilize heuristic rules to select the most possible related nodes and relations as references. Consequently, we can collect a set of edges with special relations connecting several nodes from the node candidates,

---

[4]https://www.zhixue.com/

Table 3: Evaluation results of different LLMs with CoT prompting on CARP.

| Models | Acc. | ExpAcc | Fail@where | | |
| --- | --- | --- | --- | --- | --- |
| | | | Fail@first | Fail@middle | Fail@last |
| `text-davinci-002` | 31.15 | 37.45 | 79.04 | 11.29 | 9.65 |
| `text-davinci-003` | 37.50 | 44.89 | 73.61 | 15.41 | 10.98 |
| `claude-v1.3` | 40.78 | 46.89 | 76.85 | 12.08 | 11.05 |
| `gpt-3.5-turbo` | **49.39** | **56.48** | 69.69 | 16.36 | 13.94 |

which compose the EFG of a problem. After annotation, we further design an automatic verification program to verify the completeness of the EFG and the validity of relations, and filter improper ones. Besides, we also ask teachers to check the annotated EFGs from each other, to judge if the EFG has fully covered the whole problem-solving process of the problem, and refine the incomplete ones.

## 3.2 Dataset Details

**Dataset Description.** The statistics of the CARP dataset are shown in Table 2. CARP consists of 4,886 middle school computation-intensive algebra problems, and each problem is associated with a natural language solution and an annotated EFG. Our annotated EFG explicitly depicts the step-by-step reasoning process of a math problem in a readable and concise format. On average, an EFG contains 6.0 nodes and 5.7 edges, as we only keep the expressions and conditions that lead to the final answer in the EFG. Besides, an EFG has 4.7 expression nodes on average, which are the main stem of the reasoning process and can be used for evaluating the accuracy of intermediate steps.

To solve the problems in CARP, LLMs require to iteratively perform reasoning based on math knowledge to correctly derive the intermediate math expressions, and solve it accurately. As the example in Table 1, given the conditions, a reasonable solving process should first deduce the intermediate equation $a + 2b = 3$ by substituting $x = 1$ into $ax + 2bx = 3$, and then reformulate the equation $a(y - 1) + 2b(y - 1) = 3$ to support plugging $a + 2b = 3$ into it. Such a reformulation step is not easy to reason out, and ChatGPT has made a mistake there, leading to a wrong answer.

**Evaluation Metrics.** Based on EFGs, we can evaluate the intermediate step-by-step reasoning process of LLMs. Specifically, we propose two new metrics, *i.e., ExpAcc* and *Fail@where*. ExpAcc measures the recall rate of expression nodes on the reference EFG by the generated output. Considering that a math problem may have different ways to solve it, we also regard the ancestors of a recalled expression in EFG as recalled ones, as the generated output has derived the subsequent conclusion starting from the ancestors. In this way, ExpAcc can be obtained by finding matched expression nodes in the reference EFG, then counting their ancestors and themselves as matched ones for computing the rate. We use SymPy to determine if two mathematical expressions match. Fail@where is another type of metric for analyzing where are the causes of incorrect answers, and we define three implementations, *i.e.,* Fail@first, Fail@middle, and Fail@last. The three metrics refer to the rates of making the first mistakes in the first step, middle steps, and last step (before the answer) within all generated incorrect solutions, respectively.

As shown in Table 3, we evaluate competitive LLMs [2, 77] on CARP with chain-of-thought prompt [7] and report Accuracy, ExpAcc, and Fail@where. First, all LLMs can not solve over half of the problems in CARP (under 50.0%), and the accuracy of intermediate steps is relatively low (under 57.0), indicating the difficulty of computation-intensive math reasoning. Second, all LLMs are more likely to make mistakes in the first step, while less likely in the last step. It demonstrates that LLMs are prone to fail in early steps, due to misuse of improper math knowledge or wrong calculations. Thus, careful deliberations on early steps might be promising to reduce errors of the model.

## 3.3 Tool Interfaces

As the results in Table 3 and existing work [15, 22, 10, 23]., it is hard for LLMs to solve computation-intensive math problems, especially for numerical calculation. In the real world, humans can utilize tools (*e.g.,* calculator) to avoid errors in manual work. Inspired by it, we consider augmenting LLMs with tools for handling complicated calculations. Considering the complexity of math calculation, we

devise multiple interfaces based on available tools, to provide specific and practical functionalities. All the interfaces are formulated into a unified format with detailed descriptions, to support convenient manipulation of LLMs. Concretely, we mainly utilize SymPy [78] as the tool, which is a Python library including various basic and advanced arithmetic operators. Based on it, we encapsulate three types of interfaces to help the computation of LLMs: (1) **Numerical Computation**: compute the value $v$ of an expression $e$ by calculating directly or substituting existing conditions. (2) **Equation Solving**: solve an equation or inequation $e$, or solve the system of equations or inequalities $\{e\}$. (3) **Expression Transformation**: transform an expression $e$ into the desired format $e'$.

Based on them, we devise fine-grained interfaces covering commonly used functionalities in math calculation. Details of interface definitions are listed in Appendix A. We set the name, arguments, and output formats of each interface, associated with a docstring that provides a natural language explanation for its usage. These interfaces are general to various computation-intensive math reasoning tasks, and can help LLMs perform complex computations. In addition, we also add a special interface, *think*, which can utilize the LLM to analyze existing conditions, deduce new conclusions, and create new math expressions, before or after tool manipulation. It can also help handle the cases that fail to invoke computation interfaces, where LLMs *think* to produce an output instead, to prevent the solving process from being interrupted.

### 3.4 Discussion

Our proposed CARP dataset focuses on systematically evaluating LLMs in solving computation-intensive math problems. CARP exhibits three key characteristics: First, solving problems in CARP involves multi-step reasoning with math domain knowledge and complex computations; CARP provides fine-grained interfaces to assess the ability of LLMs to manipulate tools during complex reasoning. In this context, LLMs should understand the usage of various interfaces and invoke them appropriately multiple times based on reasoning and math knowledge during the solving process. Third, evaluation metrics for intermediate reasoning steps are incorporated, based on formulated annotations. This allows for a more nuanced analysis of the multi-step reasoning performance of LLMs, in contrast to existing datasets that primarily focus on evaluating outcome accuracy [13–15]. Through these metrics, researchers can quantify the models' proficiency in the problem-solving process and gain insights for model improvement.

## 4  Approach

According to the results in Section 3.2, LLMs struggle to solve computation-intensive math problems on their own and frequently make mistakes at early reasoning steps. Inspired by the human practice of reviewing and verifying solutions, we propose a new approach that can deliberate the reasoning steps of LLMs with interfaces of tools, namely **DELI**. The overview of DELI is shown in Figure 2. In DELI, we leverage a retrieval-augmented chain-of-thought prompting strategy to initialize a step-by-step natural language solution. Then, we iterate the two-stage deliberation method that checks and refines the solution from the perspectives of natural language reasoning and tool manipulation. After multiple iterations, we can finally obtain a more reasonable solution with the answer.

### 4.1  Retrieval-Augmented Solution Initialization

As DELI focuses on deliberating over LLM-generated solutions, we aim to initialize a high-quality step-by-step solution for the given question that covers useful math knowledge and arithmetic operators. Therefore, we propose to retrieve relevant problems and solutions as the exemplars, and then utilize the chain-of-thought prompting method [7] to generate the initial solution based on them. Concretely, given a math problem $p$, we first retrieve top-$k$ relevant problems $C = \{\langle p_i, s_i \rangle\}_{i=1}^k$ from the candidate pool based on question-question matching, where the retriever can be either lexicon-based [79] or dense retrieval models [80]. Then, the retrieved problems with their associated step-by-step solutions, will be employed to compose the input prompt, to elicit LLMs for performing chain-of-thought reasoning. The pattern of the input prompt is denoted as: "*You are a helpful assistant for solving math problems in LaTeX format: [$p_1$], [$s_1$], $\cdots$, [$p_k$], [$s_k$], [$p$]*". In this way, LLMs would follow the exemplars to perform step-by-step reasoning, and can also refer to useful math knowledge from them, leading to high-quality initial solutions for deliberation. Note that such a way also supports other prompting methods to initialize solutions.

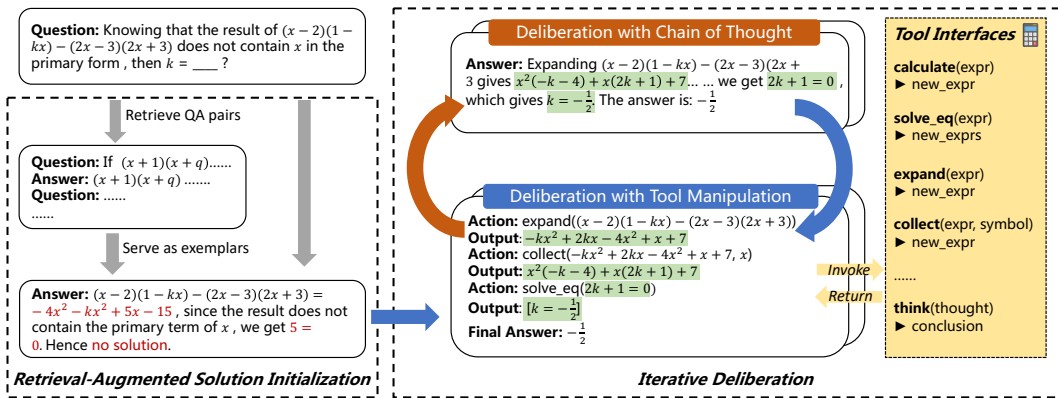

Figure 2: The overview of our DELI. DELI initializes the step-by-step solution via retrieval-augmented strategy, and then performs iterative deliberation with tool manipulation and chain of thought, respectively.

## 4.2 Iterative Deliberation

Based on the initial solution, we iterate two types of deliberation procedures, *i.e.,* deliberation with tool manipulation and deliberation with chain of thought, until reaching the stop condition. In deliberation procedures, we adopt specific in-context exemplars to guide LLMs, for checking and correcting the errors in the current solution. Next, we first introduce details of the two deliberation procedures, and then present the stop conditions.

**Deliberation with Tool Manipulation Prompting.** Since LLMs are prone to make mistakes in numerical calculation, we design the procedure of deliberation with tool manipulation, for seeking help from external tools to address it. Based on our devised interfaces in Section 3.3, we aim to rewrite the current solution into a process that orderly invokes the interfaces to produce the result. In this way, the deliberation procedure is divided into a sequence of steps, where the LLM should select the interface and then invoke it to produce the intermediate result in each step.

First, we construct an instruction introducing the goal and formats of this procedure, and details of available interfaces. For each interface, we not only list its name, arguments and description, but also provide an example to exhibit the usage, *e.g., "expand(expression: str)→ new expression: str: Expand the expression into a polynomial. For example, expand($(x + 1)^2$) → $x^2 + 2x + 1$".* Then, we demonstrate several exemplars to guide LLMs to invoke the interfaces. Each exemplar consists of four parts, *i.e.,* a question, a trial, multiple actions, and their outputs. The trial is a step-by-step solution either from the initial stage or the last iteration, which may contain errors requiring correction. Actions are a series of interface invocations derived from the trial, and outputs are the intermediate results by executing the actions, *e.g., "Action: solve_eq($2k + 1 = 0$). Output: $[k = -\frac{1}{2}]$".* Based on the instruction and exemplars, the LLM would be elicited to generate the action in formal language iteratively (*i.e.,* selecting the interface and setting its arguments), then execute it to obtain the output, until reaching the answer. To guarantee the continuity of the deliberation procedure, we set a special token after the generated action, for pausing the generation process and waiting for the result of interface invocation. In the iterative selection-then-execution process, we can deliberate the intermediate steps of the generated solution, and benefit from tool manipulation for accurate computation.

**Deliberation with Chain of Thought Prompting.** After deliberation with tools, we can obtain the solution consisting of a series of actions to invoke interfaces and their outputs. Next, we further deliberate the solution with the chain of thought to reorganize it into the natural language, which can better leverage the learned textual knowledge from LLMs to recheck it and also improve the readability.

Similarly, we also leverage an instruction with in-context exemplars to compose the input prompt. The instruction is "*You have access to both natural language problem solving processes and formal problem solving processes, but there may be errors within them. You need to learn the correct methods in order to better solve problems.*", to introduce the goal of the deliberation procedure. All

the exemplars are composed of four components, *i.e.,* a question, a given solution, the verification, and the revised solution. The given solution is the last natural language solution that is either the initial solution or the solution from the last deliberation iteration with chain of thought, and the verification is the formal language solution from the last deliberation procedure with tool interfaces. The revised solution is the result of integrating the two types of solutions into the chain-of-thought manner, where the errors and unreasonable steps have been corrected. Guided by the exemplars, LLMs would deliberate the intermediate steps from in-context solutions, and generate a new natural language solution. Besides, as there are often inconsistent intermediate computation results in the in-context solutions, we also add an instruction to elicit LLMs to trust more on the result from tool manipulation, *i.e.,* "*If the computed result in the verification differs from the computed result in the given solution, the computed result in the verification must be used as the standard*".

**Stop Conditions of Iteration.** The two deliberation procedures would be alternated multiple times, where the solution might be iteratively revised and improved. To control the cost, we set the stop conditions of the iteration process. First, once the solution of the new iteration is the same as the last one, the iteration stops, since the iteration has converged. Second, if the answers to the two deliberation procedures are consistent, we will also stop the iteration. Third, if we have reached the maximum number of iterations, the answer from the last deliberation procedure with tool manipulation will be regarded as the final answer, as the procedure can better solve computation subproblems, leading to a more accurate answer.

# 5 Experiment

## 5.1 Main Experiments

**Evaluation Datasets.** In addition to CARP, we collect 6 existing computation-intensive math datasets for evaluation, including Algebra, Prealgebra, Counting & Probability (CP) and Number Theory (NT) from MATH [15], and GK-Cloze (GKC) and SAT-Math (SAT) from AGIEval [13]. The datasets involve multi-step reasoning and computation with knowledge ranging from middle school to competition level. Details about the datasets are listed in Appendix B

**Baselines.** We compare our proposed DELI with several competitive prompting methods for LLMs. (1) *CoT prompting methods*: **Random CoT** [7] randomly selects exemplars from the training set. **Complex CoT** [81] samples the most complex problems and their solutions as exemplars. **Retrieval CoT** retrieves the most relevant problems and solutions from the training set as exemplars. (2) *Tool-augmented prompting methods*: **PAL** [26] converts the reasoning process into a Python program and executes it to get the answer. **ReAct** [32] interleave reasoning and interface invocations multiple times to get the answer, which is a basic component of DELI. (3) *Iterative prompting methods*: **Learning to Program (LP)** [74] aims to iteratively learn solutions from training sets to guide LLMs in solving similar problems based on in-context learning. **Learning to Program (LP)** [74] employs iterative learning from training sets to steer LLMs in context-based problem solving. As a variant of our framework, **Iterative CoT** integrates the existing CoT solution and self-generated feedback into a refined solution. Similarly, **Iterative ReAct** refines interface invocations using pre-existing ones and self-feedback.

**Implementation Details.** We employ OpenAI `gpt-3.5-turbo` (May 2023) API as the solver and reasoning tool and implement the computation tool based on SymPy [78]. We set the temperature to 0 and top_p to 1 for determined outputs. To retrieve similar problems, we train a sentence embedding model following SimCSE [80] to index MATH datasets and employ the BM25 algorithm [79] for the CARP dataset. The maximum number of iteration turns is set to 3 for all datasets and iterative methods. For each dataset, we specify the descriptions of interfaces that may be useful to solve the problems in prompts.

We initialize the solution with Retrieval CoT in most datasets. For GK-Cloze and SAT-Math, we initialize the solution with Random CoT, since these datasets only provide few-shot exemplars but not training sets. Following the settings in Zheng et al. [69], the initial solutions of PHP are from Complex CoT in the subsets of MATH (Algebra, Prealgebra, CP, NT), while using the same initial solutions as DELI in other datasets.

Table 4: Results on 7 computation-intensive math reasoning datasets. We copy results of LP from Guo et al. [74]. The best and second-best methods are marked in bold and underlined respectively.

| Methods | CARP | Algebra | Prealgebra | CP | NT | GKC | SAT | Avg. |
|---|---|---|---|---|---|---|---|---|
| Random CoT | 49.39 | 49.37 | 55.57 | 32.91 | 29.81 | 14.41 | 65.91 | 42.48 |
| Complex CoT | 48.06 | 51.64 | 53.73 | 32.91 | 32.22 | - | - | - |
| Retrieval CoT | 63.93 | 53.75 | 56.72 | 33.12 | 30.00 | - | - | - |
| PAL | 40.00 | 34.29 | 50.52 | 35.86 | 31.30 | 5.93 | 47.73 | 35.09 |
| ReAct | 64.11 | 54.51 | 54.53 | **41.77** | 31.67 | 16.94 | 72.27 | 48.07 |
| LP | - | 49.60 | 52.30 | 30.20 | 29.80 | - | - | - |
| PHP | 61.68 | 54.42 | 57.86 | 36.71 | **35.37** | 16.94 | 71.82 | 47.82 |
| Iterative CoT | 61.27 | 52.74 | 55.34 | 33.97 | 29.81 | 14.41 | 69.55 | 45.30 |
| Iterative ReAct | 61.17 | 53.92 | 52.12 | 37.34 | 32.22 | 15.25 | 70.00 | 46.00 |
| DELI | **73.46** | **59.65** | **58.32** | 39.03 | 33.15 | **17.80** | **74.54** | **50.85** |

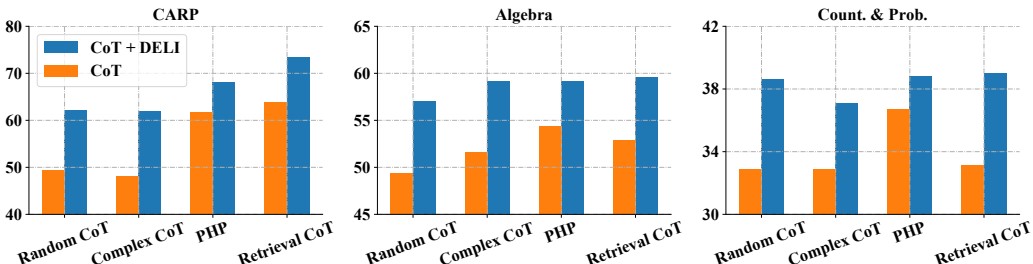

Figure 3: The results of combining DELI with existing CoT methods.

**Main Results.** Table 4 shows the results of DELI and baselines on the 7 datasets. For the comparison among CoT prompting methods, Retrieval CoT outperforms Random CoT [7] and Complex CoT [81] on average, indicating that involving relevant problems and their solutions is beneficial for answering complex math problems. Augmented with tools, ReAct [32] achieves better average performance than CoT prompting methods, showing the effectiveness of manipulating tools in solving computation-intensive math problems. Besides, ReAct can also perform better than competitive baselines, *e.g.,* PAL [26], LP [74] and PHP [69] on the benchmark. A possible reason is that ReAct adopts our devised interfaces to manipulate tools, which enables it to better deal with fine-grained computation operators. Finally, DELI performs better than all baselines in most cases. In DELI, we incorporate an iterative two-stage deliberation strategy to check and refine the generated step-by-step solutions from the LLM. Such a way is also more effective than just iterating CoT or ReAct, as it can better integrate the manipulation of tools and reasoning in natural language.

## 5.2 Analysis

**Combining with Existing CoT methods.** In DELI, we initialize a step-by-step solution by retrieving problems and solutions as exemplars. Actually, our DELI also supports other prompting methods to initialize solutions. We report the performance of combining DELI with different CoT methods on CARP, Algebra, and CP. As shown in Figure 3, DELI can greatly boost the performance of all CoT methods, which demonstrates that our DELI is general to various CoT prompting methods to fix part of their intermediate errors. Among all methods, DELI can improve the performance of Retrieval CoT a lot. It indicates that our adopted retrieval-augmented solution initialization way is more suitable for our deliberation approach.

**Impact of Iterative Deliberation Turns.** We also study how the performance of DELI changes w.r.t. the maximum iterative deliberation turns. To comprehensively investigate it, we select CARP, Algebra, and Prealgebra for evaluation, and also report the performance of Iterative CoT, Iterative ReAct, Retrieval CoT and ReAct as reference. As shown in Figure 4, the performance of DELI consistently increases w.r.t. the increasing of maximum iteration turns. For comparison, the performance of other

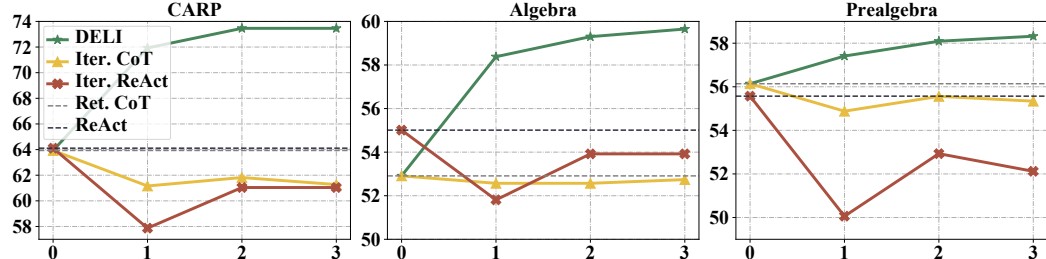

Figure 4: Accuracy of different methods w.r.t. the maximum number of iteration turns.

iterative methods does not always increase, and can even drop with an increase in maximum turns. It demonstrates that LLMs are not easy to be steered to deliberate and correct errors, and DELI is a suitable way that deliberates from the perspectives of reasoning in natural language and tool manipulation, like humans that iteratively check the solution by thinking critically and using tools.

**Evaluating Intermediate Reasoning Steps.** We evaluate the accuracy of intermediate reasoning steps for different methods via our proposed metrics ExpAcc and Fail@where on 128 challenging problems from CARP that are incorrectly answered by all the methods. As shown in Table 5, DELI achieves better ExpAcc than CoT and ReAct. Besides, the results of Fail@where show that DELI is less declined to generate completely wrong solutions (with lower Fail@first),

Table 5: ExpAcc and Fail@where on a challenging subset of CARP.

| Methods | ExpAcc | Fail@where | | |
|---|---|---|---|---|
| | | first | middle | last |
| CoT | 13.91 | 67.97 | 22.65 | 9.38 |
| ReAct | 12.58 | 66.41 | 29.69 | 3.91 |
| DELI | **18.90** | 60.16 | 25.78 | 14.06 |

and has a larger percentage of near-correct solutions (with higher Fail@last). It indicates that DELI is able to correct wrong intermediate results by the proposed iterative two-stage deliberation strategy.

# 6   Limitations

First, due to the need for quality control of the problems and the requirement of formulated annotation of intermediate reasoning steps, we retain and annotate relatively small-scale data the dataset size (at the order of thousands), future work should focus on expanding the data sources and increasing the scale of annotation. Second, to assess the tool manipulation ability of LLMs, the dataset primarily includes middle school-level math problems. Higher difficulty levels, such as high school or college-level mathematics, have not been incorporated. Future efforts can involve annotating intermediate steps for more challenging problems and designing tools for advanced math concepts. Third, the experiments conducted with OpenAI API are limited to GPT-3.5 series, as we do not acquire access to GPT-4. Consequently, the evaluations are constrained to the capabilities of GPT-3.5. To provide a more comprehensive assessment, future research can consider expanding the evaluation scope to include a wider range of base models.

# 7   Conclusion

In this paper, we proposed CARP, a computation-intensive algebra problem dataset with formulated annotation of intermediate reasoning steps for systematically evaluating LLMs in tools manipulation and math reasoning. Based on experiments in CARP, we found that popular LLMs with chain-of-thought prompting can not solve over half of the problems in CARP, and they are more likely to make mistakes in early steps, leading to wrong answers. To alleviate it, we proposed DELI, a new approach that can deliberate the intermediate reasoning steps with tool interfaces. DELI incorporated two iterative deliberation procedures to check and refine the intermediate reasoning steps of the generated step-by-step solution, from the perspectives of tool manipulation and natural language reasoning. To verify the effectiveness of DELI, we conducted extensive experiments on CARP and 6 other computation-intensive math reasoning datasets. Experimental results have shown that DELI outperforms baselines and can boost the performance of various CoT prompting methods.

## Acknowledgments and Disclosure of Funding

This work was partially supported by National Natural Science Foundation of China under Grant No. 62222215, Beijing Natural Science Foundation under Grant No. L233008 and 4222027. And this work is also partially supported by the Outstanding Innovative Talents Cultivation Funded Programs 2021 of Renmin University of China. Xin Zhao is the corresponding author.

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

# A  Interface Definition

We provide interface definitions of our tools in Table 6.

Table 6: Interface definitions of tools. Num. Comp., Eq. Solving, and Expr. Trans. refer to numerical computation, equation solving, and expression transformation, respectively.

| Category | Interface | Description |
|---|---|---|
| Num. Comp. | $calculate(e) \rightarrow v$ | Calculate the value $v$ of $e$. |
| | $substitute(e, \{c\}) \rightarrow v$ | Substitute the contextual conditions $\{c\}$ into $e$. |
| Eq. Solving | $solve\_eq(e) \rightarrow, \{e'\}$ | Solve the equation $e$ to get the solution set $\{e'\}$. |
| | $solve\_ineq(e) \rightarrow \{e'\}$ | Solve the inequation $e$ to get the solution set $\{e'\}$. |
| | $solve\_multi\_eq(\{e\}) \rightarrow \{e'\}$ | Solve the system of equations to get the solution set $\{e'\}$. |
| | $solve\_multi\_ineq(\{e\})$ $\rightarrow \{e'\}$ | Solve the system of inequations to get the solution set $\{e'\}$. |
| | $partial\_solve(e, u) \rightarrow \{e'\}$ | Solve the equation $e$ assuming that $u$ is an unknown to get the solution set $\{e'\}$. |
| Expr. Trans. | $expand(e) \rightarrow e'$ | Expand $e$ to get $e'$. |
| | $factor(e) \rightarrow e'$ | Factorize $e$ to get $e'$. |
| | $collect(e, x) \rightarrow e'$ | Collect $e$ based on the symbol $x$ to get $e'$. |
| | $complete\_the\_square(e) \rightarrow e'$ | Complete the square of $e$ to get $e'$ |
| Thinking | $think(l) \rightarrow l'$ | Draw a conclusion $l'$ based on the free thought $l$. |

# B  Details of Evaluated Datasets

We provide details of evaluating datasets in Table 7.

Table 7: Basic information about datasets in evaluated datasets. MS and HS refer to "middle school" and "high school", respectively.

| Dataset | Source | Language | Domain | Difficulty | Train | Test |
|---|---|---|---|---|---|---|
| CARP | Ours | Chinese | Algebra | MS | 3,410 | 976 |
| Algebra | MATH | English | Algebra | HS | 1,744 | 1,187 |
| Prealgebra | MATH | English | Algebra | HS | 1,205 | 871 |
| Count. & Prob. | MATH | English | Probability | HS | 771 | 474 |
| Num. Theory | MATH | English | Number Theory | HS | 869 | 540 |
| GK-Cloze | AGIEval | Chinese | Mixture | HS | - | 220 |
| SAT-Math | AGIEval | English | Mixture | HS | - | 351 |

# C  Case Study

To better present the process of DELI, we provide a case study that shows the solving process of DELI on CARP, which is shown in Figure 5. We also report the solution of Retrieval CoT and ReAct in the figure. It is noted that the solution of Retrieval CoT is also the initial solution in DELI.

First, both Retrieval CoT and ReAct make minor mistakes in the solving process. Although following the correct solving idea from relevant solutions, Retrieval CoT struggles with expanding the expression $(x-2)(1-kx) - (2x-3)(2x+3)$, leading to an incorrect intermediate result. Besides, ReAct fails at understanding the condition *the expression does not contain the primary form of $x$*, thus collecting the expression according to a wrong term $x^2$. Therefore, both CoT and ReAct can not solve the case individually due to the challenges of computations and reasoning.

DELI iterates over the existing solutions. In deliberation with tool manipulation, the model reviews the existing natural language solution, and invokes interfaces based on the ideas therein. In this case,

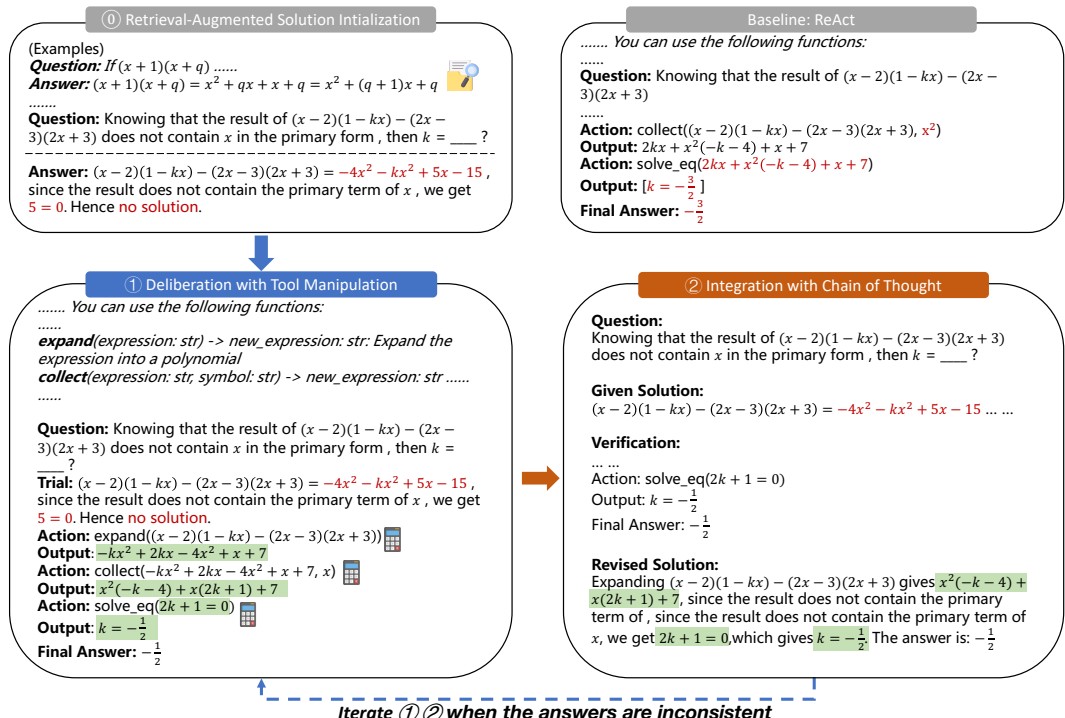

Figure 5: Case study of our method with baselines on the CARP dataset. The case is translated into English.

the model invokes interfaces *expand* and *collect* in a row to get the correct expanded expression $(x − 2)(1 − kx) − (2x − 3)(2x + 3) = −kx^2 + 2kx − 4x^2 + 7$ with the help of tools. Then, the model solves the equation derived from the expanded expression and gets the correct answer.

In deliberation with chain of thought, the model reviews both natural language and formal language solutions from the previous iteration and generates a revised CoT solution, which fixes the computation error in the original CoT solution according to interface invocations. In this case, due to the consistent answers between the revised CoT solution and the previous ReAct solution, the iteration terminates. In general cases, the iteration continues until solutions converge or the answer is consistent, or the maximum number of iterations is reached.

## D   Prompts for Two-Stage Deliberation

We list the prompts for two-stage deliberation on CARP. The prompts are translated into English.

Listing 1: Prompt for Deliberation with Tool Interfaces.

```
You are ChatGPT, a math problem solver equipped with multiple functions to tackle
    various math problems. While you may have access to existing problem-solving
    processes, there is a possibility of errors. Therefore, you need to learn the
    correct approaches to solve problems more efficiently. You can use the
    following functions:

calculate(expression: str) -> new_expression: str: Calculate the value of the
    expression and return it as a string. For example, calculate("34 * 2") -> "68".
solve_eq(expression: str) -> new_expressions: list: Solve the equation expression
    and return the result as a list. For example, solve_eq("3 x + 4 = 1") -> ["x =
    -1"].
solve_ineq(expression: str) -> new_expression: str: Solve the inequality expression
    and return the result as a string. For example, solve_ineq("3 x + 4 < 1") -> "x
    < -1".
```

```
solve_multi_eq(expressions: list) -> new_expressions: dict: Solve the system of
    equations given by the list of expressions and return the result as a
    dictionary. For example, solve_multi_eq(["x + y = 2", "x - 2 y = -7"]) -> {"x":
    ["x = -1"], "y": ["y = 3"]}.
solve_multi_ineq(expressions: list) -> new_expression: str: Solve the system of
    inequalities given by the list of expressions and return the result as a string.
    For example, solve_multi_ineq(["x \le 2", "x \le -7"]) -> "x \le -7".
substitute(expression: str, conditions: list[str]) -> new_expression: str:
    Substitute the contextual conditions in the list into the expression and return
    the result. For example, substitute("3 x + 4", ["x = 1"]) -> "7".
expand(expression: str) -> new_expression: str: Expand the expression into a
    polynomial. For example, expand("(x + 1) ^ 2") -> "x ^ 2 + 2x + 1"
factor(expression: str) -> new_expression: str: Factorize the polynomial. For
    example, factor("x ^ 2 + 2x + 1") -> "(x + 1) ^ 2"
collect(expression: str, symbol: str) -> new_expression: str: Collect the
    coefficients of the corresponding powers according to the given symbol. For
    example, collect("a x - 5 a + x ^ { 2 } - 5 x", "x") -> "- 5 a + x ^ { 2 } + x
    ( a - 5 )"
partial_solve(expression: str, symbol: str) -> new_expression: str: Let the given
    symbol be the unknown variable and solve the linear equation expression with
    one variable. For example, partial_solve("x + 3 y - 3 = 0", "x") -> "x = - 3 y +
     3"
think(thought: str) -> conclusion: str: Generate new conclusions based on natural
    language description thought. Think should only be called when the above
    functions are not applicable. For example, think("\sqrt{x-8} the expression
    inside the root is always greater than or equal to 0") -> "x-8\\ge0"

To use ChatGPT, simply provide a mathematical problem or question in LaTeX format.
    You can use any of the above functions to help solve the problem. Please follow
     the following format:

Question: The input question you must answer. This appears only once.
Trial: The problem-solving approach that can be referred to. It may contain errors,
    you can refer to the correct part in it.
Action: A function call, which should be one of the mentioned functions with
    arguments. You must only call one function in one Action.
Output: The result of the action. Every Action must be immediately followed by one
    and only one Output.
... (This Action/Output cycle can repeat N times.)
Final Answer: The final answer to the original input question. The answer should be
    numerical or LaTeX math expression. Do not use natural language in the answer
```

Listing 2: Prompt for Deliberation with Chain of Thought.

```
You are ChatGPT, a mathematical problem solver equipped with multiple functions for
    solving mathematical problems. You have access to both natural language problem
     solving processes and formal problem solving processes, but there may be
    errors within them. You need to learn the correct methods in order to better
    solve problems. You can use the following functions:

calculate(expression: str) -> new_expression: str: Calculate the value of the
    expression and return it as a string. For example, calculate("34 * 2") -> "68".
solve_eq(expression: str) -> new_expressions: list: Solve the equation expression
    and return the result as a list. For example, solve_eq("3 x + 4 = 1") -> ["x =
    -1"].
solve_ineq(expression: str) -> new_expression: str: Solve the inequality expression
    and return the result as a string. For example, solve_ineq("3 x + 4 < 1") -> "x
     < -1".
solve_multi_eq(expressions: list) -> new_expressions: dict: Solve the system of
    equations given by the list of expressions and return the result as a
    dictionary. For example, solve_multi_eq(["x + y = 2", "x - 2 y = -7"]) -> {"x":
    ["x = -1"], "y": ["y = 3"]}.
solve_multi_ineq(expressions: list) -> new_expression: str: Solve the system of
    inequalities given by the list of expressions and return the result as a string.
     For example, solve_multi_ineq(["x \le 2", "x \le -7"]) -> "x \le -7".
```

```
substitute(expression: str, conditions: list[str]) -> new_expression: str:
    Substitute the contextual conditions in the list into the expression and return
     the result. For example, substitute("3 x + 4", ["x = 1"]) -> "7".
expand(expression: str) -> new_expression: str: Expand the expression into a
    polynomial. For example, expand("(x + 1) ^ 2") -> "x ^ 2 + 2x + 1"
factor(expression: str) -> new_expression: str: Factorize the polynomial. For
    example, factor("x ^ 2 + 2x + 1") -> "(x + 1) ^ 2"
collect(expression: str, symbol: str) -> new_expression: str: Collect the
    coefficients of the corresponding powers according to the given symbol. For
    example, collect("a x - 5 a + x ^ { 2 } - 5 x", "x") -> "- 5 a + x ^ { 2 } + x
    ( a - 5 )"
partial_solve(expression: str, symbol: str) -> new_expression: str: Let the given
    symbol be the unknown variable and solve the linear equation expression with
    one variable. For example, partial_solve("x + 3 y - 3 = 0", "x) -> "x = - 3 y +
     3"
think(thought: str) -> conclusion: str: Generate new conclusions based on natural
    language description thought. Think should only be called when the above
    functions are not applicable. For example, think("\sqrt{x-8} the expression
    inside the root is always greater than or equal to 0") -> "x-8\\ge0"

Follow this format:

‘‘‘

Question:
The input question that you must answer. It appears only once.

Given Solution:
A natural language solution that can be used as a reference, which may contain
    errors. You can refer to the correct ideas in it.

Verification: Transform the original solution into a verification process that uses
    functions, corrects any computational errors, and simplifies the process.
Action: A function call, which must be one of the functions mentioned above and
    include parameters. You can only call one function in an Action.
Output: The result of an Action. Each Action must have one and only one Output
    following it.
(Action/Output can be repeated any number of times...)
Final Answer: The ultimate solution to the original input problem.

Revise the given solution based on the verification process:
Revise the original solution based on the computed result in the verification
    process. If the computed result in the verification process differs from the
    computed result in the original solution, the computed result in the
    verification process must be used as the standard.
‘‘‘
```

# E    Annotation Platform

We build an annotation platform for crowdworkers to annotate EFGs in CARP. A screenshot of the platform is provided in Figure 6. First, given output nodes, crowdworkers are required to find the nodes that have directed edges to the output nodes. Second, crowdworkers are required to annotate relations between each node pair found in the first stage. Finally, the crowdworkers should review the annotation, and remove irrelevant nodes that do not contribute to the reasoning process. Besides, if the problem or solution has errors, the crowdworkers should annotate the error types and skip them.

**ID**

ed58e4bc-6b16-49db-8fed-d63b634ca76f

**Problem**

The solution to the equation $ax + 2bx = 3$ is $x = 1$, then the solution to the equation $a(y-1) + 2b(y-1) = 3$ is ?

**Answer**

y = 2

**Solution**

From the question we have: $a + 2b = 3$, $a(y-1) + 2b(y-1) = 3$. Rectifying gives $(a + 2b)y - (a + 2b) = 3$, i.e., $3y - 3 = 3$, therefore $y = 2$。

| ... ▾ | | Supplement |
|---|---|---|

| Input Expression Nodes | Input Natural Language Nodes | Output Expression Nodes | Confirm | Delete |
|---|---|---|---|---|
| $ax + 2bx = 3, x = 1$ ▾ | Nothing selected ▾ | $a + 2b = 3$ | Confirm | Delete |
| $a(y-1) + 2b(y-1) = 3, a + 2b = 3$ ▾ | Nothing selected ▾ | $(a + 2b)y - (a + 2b) = 3$ | Confirm | Delete |
| $a + 2b = 3, (a + 2b)y - (a + 2b) = 3$ ▾ | Nothing selected ▾ | $3y - 3 = 3$ | Confirm | Delete |
| $3y - 3 = 3$ ▾ | Nothing selected ▾ | $y = 2$ | Confirm | Delete |

| Input Nodes | Relation Types | Output Nodes |
|---|---|---|
| $ax + 2bx = 3$ | substituted expr ▾ | $a + 2b = 3$ |
| $x = 1$ | substitution ▾ | $a + 2b = 3$ |
| $a(y-1) + 2b(y-1) = 3$ | factored expr ▾ | $(a + 2b)y - (a + 2b) = 3$ |
| $a + 2b = 3$ | common factor ▾ | $(a + 2b)y - (a + 2b) = 3$ |
| $(a + 2b)y - (a + 2b) = 3$ | substituted expr ▾ | $3y - 3 = 3$ |
| $a + 2b = 3$ | substitution ▾ | $3y - 3 = 3$ |
| $3y - 3 = 3$ | equation ▾ | $y = 2$ |

Modify

Reset

**Issues**

☐ Missing neccessary output nodes
☐ Missing neccessary input nodes
☐ Not algebra
☐ Exist erros in the problem or the solution
☐ Can not represent the solution with EFG
☐ Automatic verification does not meet expectations

Report Issues

Home

Figure 6: A screenshot of the EFG annotation platform. The platform has been translated into English.

