# OpenReview forum: "Evaluating and Improving Tool-Augmented Computation-Intensive Math Reasoning"
_NeurIPS.cc/2023/Track/Datasets_and_Benchmarks — NeurIPS 2023 Datasets and Benchmarks Poster_

### Official Review · Reviewer_73Fj · 2023-07-04
**Incomprehensive evaluation of tool manipulation ability**

**Rating:** 4
**Confidence:** 5
**Correctness:** Yes
**Clarity:** Yes

**Strengths:**

1. The paper proposes a new dataset called CARP that consists of 4,886 computation-intensive algebra problems with formal annotations of all intermediate steps, which enables researchers to test the accuracy of intermediate reasoning steps for analyzing the errors of LLMs.

2. The paper introduces a new evaluation metric called Intermediate Reasoning Accuracy (IRA) that measures the accuracy of intermediate reasoning steps in LLMs' solutions to math problems.

3. The paper presents experimental results on four LLMs with chain-of-thought (CoT) prompting on CARP, which show that CoT prompting can significantly improve LLMs' math reasoning performance.

4. The paper proposes a new approach called DELI (Deliberation with Exemplar-based Learning and Interface) that can deliberate the reasoning steps of LLMs with a tool interface.

**Additional Feedback:**

None

**Documentation:**

Yes

**Limitations:**

This paper doesn't discuss the limitations of this work.

**Opportunities For Improvement:**

1. Unclear motivation. What exactly are you trying to measure? The ability to use tools? If so, I do not see the contribution of creating a dataset that contains annotations for intermediate steps to better quantify the tool manipulation performance.

2. Based on the first point, if this work motivates to measure the tool manipulation ability, the benchmark is far from comprehensive. First, only three tools are considered in this work (e.g., equation solving), which is insufficient compared to some contemporary work on measuring tool manipulation ability. Second, the widely-used interpreter for math reasoning is also not included in the toolset. Third, the evaluation is only focusing on math reasoning and algebra problem, lacking diversity compared to contemporary work on measuring tool manipulation ability across several different domains & various APIs.

3. For the evaluation benchmark, is this annotated intermediate reasoning step the only way to derive the final answer? If I understand correctly, there are multiple ways or multiple paths to solve an algebra problem. How do you ensure that the question has only one path or one golden solution?  If it can not be verified, the conclusion: "LLMs are all prone to make mistakes at the early steps of the solution, leading to incorrect answers" is also invalid.

4. Although there are no explicit numbers in the paper, I suspect that the computational cost of the proposed method is significantly higher than the baseline methods. Also, I do not see too much novelty from the proposed method, compared to the React framework. The core parts of them both contain two stages, where the former stage requires LLMs to think/reason, and the latter stage requires LLMs to manipulate tools/make actions.

**Relation To Prior Work:**

Yes

**Summary And Contributions:**

This paper is about evaluating and improving tool-augmented computation-intensive math reasoning using large language models (LLMs). It discusses the challenges of developing datasets and evaluation metrics for LLMs' math reasoning abilities and proposes a new dataset called CARP (Computation-Intensive ARithmetic Problems) that addresses these challenges. The paper also presents experimental results on four LLMs with chain-of-thought prompting on CARP and discusses the implications of these findings for future research on improving LLMs' math reasoning capabilities.

---

> ### Author Response · Authors · 2023-08-21
> **Response to the Concerns of Reviewer [Part 1]**
>
> Thanks for your insightful suggestions and we have listed our response to the concerns as follows. If you still have any other questions, please do not hesitate to tell us. We will continue to try our best to answer for you.
>
> **1. Unclear motivation. What exactly are you trying to measure? The ability to use tools? If so, I do not see the contribution of creating a dataset that contains annotations for intermediate steps to better quantify the tool manipulation performance.**
>
> In this paper, we aim to evaluate the performance of LLMs with the help of tool manipulation on the computation-intensive mathematical reasoning task.
>
> Currently, there are few existing datasets that focus on the computation-intensive mathematical reasoning task, and they are mostly unable to evaluate the correctness of the intermediate steps. Thus, we aim to provide a high-quality dataset CARP, that can enrich the data resource in this task, and also provide formulated annotations about the intermediate computation steps, which is useful to study the mistakes of LLMs in the intermediate steps. Based on these annotations and the preliminary experiment, we obtain an interesting finding that LLMs mostly make the first mistake in early steps (69% errors in the first step). Once LLMs make mistakes in the early step, the following steps will be meaningless reasoning processes but just accumulate the errors to the incorrect final answer.
>
> Such a finding inspires us to make the following efforts for new evaluating metrics and new approaches for improving LLMs. First, we design new evaluation metrics for evaluating the intermediate reasoning process, including ExpAcc and Fail@where. ExpAcc can rely on our annotations about the intermediate steps to evaluate the Accuracy of intermediate results, it is a useful complementary evaluation metric for the commonly-used Accuracy of the final answer. Fail@first/middle/last denotes the rate of making the "first mistakes" in the first/middle/last step among all instances with incorrect answers, which is useful to analyze and diagnose the characteristics of the errors for the model. Second, we propose the approach DELI, a prompting method that can deliberate and refine the generated solution, especially the intermediate steps. In DELI, we consider two important aspects of the solution, i.e., natural language reasoning and tool manipulation, and iterate the two perspectives of deliberations to gradually improve the quality of the solution.
>
> **2. Only three tools are considered in this work (e.g., equation solving), which is insufficient compared to some contemporary work on measuring tool manipulation ability.**
>
> As shown in Table 6 of the supplemental material, we have considered 12 interfaces specially for different functionalities, and they can well cover most of the requirements for our focused computation-intensive mathematical reasoning task, e.g., numerical computation and expression transformation. In this work, as the usage of external tools is mainly to improve the capability of LLMs for handling complex computations, it is not necessary to adopt so many tools that may also increase the maintenance overhead.
>
> Besides, it is easy to extend the utilization of new tools in our method, as we have provided a flexible framework that researchers can just add new interfaces for tools as in Table 6. Concretely, the detailed implementation and tools can be black boxes, and only the arguments and functionality should be added to our prompts to help guide the LLM. To verify it, we add three new tools with corresponding interfaces into our approach (i.e., prime factorization, modulo operations, and number base conversion) as follows:
> > prime_factors(number: str) -> new_expression: str: Generate a prime factorization expression for a given number. E.g., prime_factors("360") -> "2 ^ 3 \times 3 ^ 2 \times 5"
> > mod(dividend: str, divisor: str) -> new_expression: str: Calculates the remainder of dividing the dividend by the divisor. E.g., mod("9", "6") -> "3".
> > base_conversion(number: str, to_base: str) -> Converts a number from one base to another. E.g., base_conversion("11_2", "10") -> "3_{10}"
>
> Then, we report the experimental results of our approach on the Number Theory dataset in the following table. We can see that extending the toolset with appropriate interfaces can further improve the performance of our approach.
>
>
> \\begin{array} {|l|l|c|}
> \\hline
> \\textrm{Model} & \\textrm{Extended Interfaces} & \\textrm{DELI}  \\\\
> \\hline
> \\textrm{gpt-3.5-turbo (Aug.)} & \\textrm{Original} & 36.22  \\\\
> & \\textrm{prime\\_factors} & 36.85  \\\\
> & \\textrm{mod, base\\_conversion} & 37.22  \\\\
> & \\textrm{All} & \\textbf{37.41}  \\\\ \\hline
> \\end{array}

---

> ### Author Response · Authors · 2023-08-21
> **Response to the Concerns of Reviewer [Part 2]**
>
> **3. The widely-used interpreter for math reasoning is also not included in the toolset.**
>
> For the reviewer mentioned the code interpreter, our approach has indeed adopted it, and all our interfaces are designed as the format of Python function, which are executed for the code interpreter to fulfill the functionality. In this way, we think the code interpreter is not the external tool but the intrinsic implementation of the interfaces to manipulate tools. And such a way can decompose the natural language reasoning and the invoking of tools, where LLMs can freely switch from both operations with proper prompt guidance. Our experimental results in Table 4 also show that our approach with the above way using an interpreter can perform better than the PAL method that regards the code interpreter as the tool.
>
> **4. The evaluation is only focusing on math reasoning and algebra problem, lacking diversity compared to contemporary work on measuring tool manipulation ability across several different domains & various APIs.**
>
> In this work, we mainly focus on the computation-intensive math reasoning task, and conduct more experiments on it. But our proposed approach, DELI is also general to various tasks that require to perform natural language reasoning and tool manipulation. To verify it, we add an experiment on HotpotQA, a challenging multi-hop knowledge reasoning dataset requiring models to search for evidence and perform reasoning over multiple supporting documents to reach the answer. We implement the baseline ReAct and DELI on gpt-3.5-turbo (Aug.), and follow existing work [1] to provide two tool interfaces:
> > (1) Search[entity], which searches the exact entity on Wikipedia and returns the first paragraph if it exists. If not, it will return some similar entities to search.
> > (2) Lookup[keyword], which returns the next sentence containing keyword in the current passage.
>
> As shown in the following table, DELI can also improve the performance of baseline methods in this task with new tool interfaces, indicating its generality in other domains and APIs.
>
> \\begin{array} {|l|c|}
> \\hline
> \\textrm{Model} & \\textrm{EM} \\\\
> \\hline
> \\textrm{ReAct} & 23.80 \\\\ \\hline
> \\textrm{Random CoT} & 30.27  \\\\  \\hline
> \\textrm{DELI} & \\textbf{32.53} \\\\
> \\end{array}
>
>
> **5. For the evaluation benchmark, is this annotated intermediate reasoning step the only way to derive the final answer? If I understand correctly, there are multiple ways or multiple paths to solve an algebra problem. How do you ensure that the question has only one path or one golden solution? If it can not be verified, the conclusion: "LLMs are all prone to make mistakes at the early steps of the solution, leading to incorrect answers" is also invalid.**
>
> Actually, based on the annotations in CARP, we are able to handle the multiple reasonable ways to solve the problem. As we guarantee the final answer of each question is unique, we find that all the possible reasonable paths just fork in the intermediate steps and will merge in the later, to reach the unique answer. Thus, we check the correctness of the reasoning steps from back to front.
>
> For example, given the problem ``Knowing that the fractional equation $-\\frac { 3 } { m + 6 } = \\frac { m } { x }$ with respect to $x$ has the same solution as the fractional equation $\frac { 3 } { 2 x } = \\frac { 1 } { x - 1 }$, then the value of $m ^ { 2 } - 2 m$ is ____ ?'', if the i-th reasoning step matched the annotated steps (e.g., $m=3$), its front steps, i.e., from the 1-th to (i-1)-th ones, are all regared as the corrrect ones, whatever they differ from the annotated ones (e.g., directly solving the solution $\frac { 3 } { 2 x } = \\frac { 1 } { x - 1 }$ to get $x=3$ or substituting $x=\frac{m(m+6)}{3}$ into $\frac { 3 } { 2 x } = \\frac { 1 } { x - 1 }$). It can fairly assign the same scores for multiple reasonable but different solutions.
>
> Considering the above evaluation method, Fail@first means that LLMs can not derive any key points correctly, even the key points in the earliest steps. Since LLMs exhibit high Fail@first (e.g. 69% for CoT based on the powerful gpt-3.5-turbo), we can draw a conclusion that LLMs are prone to make mistakes at the early steps of the solution.

---

> ### Author Response · Authors · 2023-08-21
> **Response to the Concerns of Reviewer [Part 3]**
>
> **6. Although there are no explicit numbers in the paper, I suspect that the computational cost of the proposed method is significantly higher than the baseline methods.**
>
> In DELI, the additional inference overhead mainly derives from the iterations of the two deliberation processes. To reduce the overhead, in addition to the maximum turns, we also design the stop condition that once the result before and after the deliberation iteration is not changed (indicating it has converged), we stop the iterations. We have compared existing iterative baseline methods (such as Learning to Program, PHP, and Iterative CoT/ReAct) in Table 4. In practice, we find that our DELI only requires less than 4 iterations on average. We report the average turn numbers of all evaluated datasets as follows:
>
>
> \begin{array} {|l|c|c|c|c|c|c|c|}
> \hline
> \textrm{Method} & \textrm{CARP} & \textrm{Algebra}   & \textrm{Prealgebra} & \textrm{CP}  & \textrm{NT}  & \textrm{GKC} & \textrm{SAT} \\\\
> \\hline
> \textrm{DELI} & 3.52 & 3.49 & 3.27 & 3.29 & 3.32 & 3.72 & 3.97 \\
> \end{array}
>
> Besides, we also add experiments based on gpt-3.5-turbo (Aug.) to compare our method with Self-Consistency of Retrieval CoT and ReAct in the following table, where we follow PAL that sets the temperature as 0.7, and set the number of predicted solutions as 4 for a fair comparison under the same inference overhead.
>
> \\begin{array} {|l|c|c|c|c|c|c|c|}
> \\hline
> \\textrm{Method} & \\textrm{CARP} & \\textrm{Algebra}   & \\textrm{Prealgebra} & \\textrm{CP}  & \\textrm{NT}  & \\textrm{GKC} & \\textrm{SAT} & \\textrm{Avg}. \\\\
> \\hline
> \\textrm{Retrieval CoT SC n=4} & 70.49 & 65.37 & \\textbf{62.69} & 37.34 & \\textbf{36.48} & 11.86 & 75.00 & 51.32 \\\\ \\hline
> \\textrm{ReAct SC n=4} & 64.56 & 56.11 & 56.83 & 38.4 & 30.74 & 17.8 & 65.00 & 47.06 \\\\ \\hline
> \\textrm{DELI} & \\textbf{75.34} & \\textbf{66.29} & 62.21 & \\textbf{41.42} & 36.22 & \\textbf{18.92} & \\textbf{75.75} & \\textbf{53.74} \\\\ \\hline
> \\end{array}
>
> As shown in the table, our DELI outperforms the methods using self-consistency on most datasets. The reason is that the computation-intensive task is very difficult and the generated reasoning paths of LLMs are hard to consistently hit the correct answer, especially since they are easy to make mistakes in the early steps leading to diverse wrong answers. As a comparison, our DELI can deliberate and refine the generated solution from two different perspectives, which is more suitable to handle the error-prone computation-intensive task.
>
> **7. I do not see too much novelty from the proposed method, compared to the React framework. The core parts of them both contain two stages, where the former stage requires LLMs to think/reason, and the latter stage requires LLMs to manipulate tools/make actions.**
>
> The major difference between ReAct and DELI is that DELI incorporates the deliberation strategy to rethink and refine the generated solutions from two perspectives, including natural language reasoning and tool manipulation. In contrast, ReAct adopts the one-pass generation manner that would not revise the generated past steps, which makes it unable to fix the errors in past steps. As shown in Table 5, the performance of DELI on ExpAcc is better than ReAct, indicating its effectiveness to refine the errors in intermediate steps.
> Besides, as our DELI also considers the chain-of-thought prompting strategy to generate the natural language solution, it provides a new perspective to check and refine the possible wrong steps in the solution generated by tool manipulation, using the knowledge of natural language reasoning with the LLMs. Therefore, as shown in Table 4, our DELI consistently outperforms ReAct in most datasets.
>
> **8. This paper doesn't discuss the limitations of this work.**
>
> The limitation of our work is listed in the supplemental material (page 16). Following your suggestion, we will update the section in the main paper.

---

> ### Author Response · Authors · 2023-08-26
> **Gentle Reminder for the Discussion**
>
> Dear Reviewer 73Fj,
>
> Thanks for your careful reading and insightful feedback of our paper. We have clarified the motivation of our dataset, and the novelty of our method compared to ReAct. Besides, we have added the discussion and experiment of the sufficiency and scalability of our tool interfaces for computation-intensive math reasoning. We have also discussed the relationship between our proposed interfaces and the code interpreter. We have also added a discussion of the applicability of our evaluation method for multiple possible reasoning paths. We have added an experiment comparing our method with Self-Consistency.  We would like to know whether you find our response satisfactory, or if there are more questions that we could clarify. Since the rebuttal stage is coming to an end, we are more than happy to hear your comments and address any of your further concerns during the remaining time.
>
> Best,
>
> Authors

---

> > ### Comment · Reviewer_73Fj · 2023-08-28
> >
> > Dear authors,
> >
> > Thanks for your detailed responses. I'm sorry that I did not provide in-time responses because of some personal issues. I have now carefully read your responses and listed my comments as follows:
> >
> > (1) I now have a clear understanding of the motivation of this work. Please consider making it clear in the introduction/abs.
> >
> > (2,3) Thanks for making it clear to me.
> >
> >  (4) I don't think the provided arguments have addressed my concerns. So, to make it clear, my concern is that the created benchmark is only focusing on math reasoning and algebra problems. Compared to the contemporary benchmarks that motivate to measure the tool manipulation ability,  the created benchmark is too monotonous.
> >
> > (5) If I understand correctly, your argument is if we check that the i-th step is correct, then all steps before i-th are correct. I'm not convinced because we can possibly derive a correct answer with incorrect middle steps. It doesn't happen very often but it does happen. Could you elaborate more on it?
> >
> > (6) I'm sorry but I do not see too much useful information in the provided results. First, using the iteration number to quantify the inference cost is not reasonable. Maybe the output token number is a more effective indicator. Second, if I understand correctly, the second table compares the performance of DELI with ReAct. To argue that the method is infernece-efficient, the authors should compare the inference cost of DELI with ReAct.
> >
> > (7) Thanks for elaborating more on the proposed method for me. However, I still think the novelty of the proposed method is limited. This approach is more about trading the time spent on inference for the performance improvement. In addition, Whether it is worth it remains unproven, considering (6).

---

> > > ### Author Response · Authors · 2023-08-29
> > > **Reponse to the Concerns of Reviewer [Part 1]**
> > >
> > > Thanks for your careful reading and insightful further feedback. We have listed our response to the remained concerns as follows. If you still have any other questions, please do not hesitate to tell us. We will continue to try our best to answer for you.
> > >
> > > **1. I now have a clear understanding of the motivation of this work. Please consider making it clear in the introduction/abs.**
> > >
> > > We thank the reviewer for the feedback, we will revise the introduction and abstract part as the reviewer suggested, to make it more clear.
> > >
> > > **4. The created benchmark is only focusing on math reasoning and algebra problems. Compared to the contemporary benchmarks that motivate to measure the tool manipulation ability, the created benchmark is too monotonous.**
> > >
> > > Similar to the OpenAI's research work "Let's verify step by step" [1], one major motivation of this study is to enable the evaluation of intermediate steps for reasoning. We believe that math is a good domain for creating such an evaluation dataset, because it has more formal and strict reasoning rules/constraints, thus having a clear reasoning procedure. As a comparison, the reasoning process of the tasks in other domains may not be explicitly clear or difficult to annotate by humans (making the step-by-step evaluation infeasible).
> > >
> > > Although there are several datasets on tool manipulation similar to our datasets, they are almost synthesized by LLMs, which may not be realistic or well-verified. In addition, though only the math domain is considered, we argue that the math reasoning performance of a LLM can actually reflect its comprehensive ability. For example, on many benchmarks, math reasoning is still among the most difficult evaluation task, the LLMs that have a high math reasoning performance often has a strong comprehensive ability. This is because that the underlying math reasoning ability is emergent by pre-training unlabeled text corpus, including both natural language and math corpus. Only when the models well understand the semantics of natural language and learn from various domains (since reasoning is an abstract ability, and it occurs explicitly and implicitly in different domains or tasks), they can grasp a good reasoning ability.  Therefore, our dataset provides a good testbed for evaluating advanced tool manipulation in complex reasoning scenarios.
> > >
> > > [1] Lightman H, et al. Let's Verify Step by Step. arXiv preprint arXiv:2305.20050.
> > >
> > > **5. I'm not convinced because we can possibly derive a correct answer with incorrect middle steps. It doesn't happen very often but it does happen. Could you elaborate more on it?**
> > >
> > > Thanks for the insightful comment. However, we believe that it is a fundamental issue in all the reasoning-related datasets (e.g., CoT evaluation datasets). To the best of our knowledge, there is still no standard approach to fully mitigate this risk in the literature. For our dataset, we were aware of this issue and tried very hard to reduce such cases. The solutions that we have taken are listed as follows:
> > >
> > > 1. We tend to include problems whose results (or intermediate results) are a large value range (e.g., fractions or math expressions), which significantly reduces the success rate for an easy guess. In other words, the LLMs have to make specific reasoning to approach the correct answer, in most cases (including intermediate results).
> > > 2. We have included tools in these reasoning intermediate steps, and this can largely reduce the possibility of deriving incorrect answers with correct procedures or correct answers based on incorrect procedures. Based on our empirical observation, incorrect result computation is an important error cause for the reviewer's concern, and tools can effectively alleviate this issue.
> > > 3. Our reasoning procedure is organized in graphs but not simple chains. A benefit is that it creates a complex reasoning procedure that is more difficult for random guesses (chain-like procedures are easier to occur in the cases that the reviewer raised). It is because LLMs have to combine the information from multiple branches and it is less likely to be affected by the incorrectness of a single path.
> > >
> > > Considering the reviewer's concern, we have conducted empirical experiments to quantify such possibilities. We sampled 100 questions along with solutions from both Retrieval CoT and DELI, and manually examined whether there are any incorrect steps that lead to a correct intermediate or final answer in all reasoning steps. As a result, we only encountered 2 steps (out of 393 steps) of this situation in the case of Retrieval CoT and 1 step (out of 367 steps)  in the case of DELI. The key finding is that the case that the reviewer is concerned about has a very small probability of occurring, being less than 1%.
> > >
> > > We totally agree that it is good to include more explanation or analysis in our dataset on this issue as the reviewer suggested. Our empirical analysis has further verified the reliability of this evaluation dataset.

---

> > > ### Author Response · Authors · 2023-08-29
> > > **Reponse to the Concerns of Reviewer [Part 2]**
> > >
> > > **6. Using the iteration number to quantify the inference cost is not reasonable. Maybe the output token number is a more effective indicator. Second, if I understand correctly, the second table compares the performance of DELI with ReAct. To argue that the method is infernece-efficient, the authors should compare the inference cost of DELI with ReAct.**
> > >
> > > In the second table of the previous response, we report the result of DELI and baselines (Retrieval CoT and ReAct) augmented with Self-Consistency [1]. We set the turn number of Self-Consistency to 4, which is similar to the average turn number of DELI. Results show that DELI outperforms baselines on most datasets.
> > >
> > > Following your suggestion, we further report the average output token number of DELI and baselines with Self-Consistency based on gpt-3.5-turbo as follows:
> > >
> > > \begin{array} {|l|c|c|c|c|c|c|c|c|}
> > > \hline
> > > \textrm{Method} & \textrm{CARP} & \textrm{Algebra}   & \textrm{Prealgebra} & \textrm{CP}  & \textrm{NT}  & \textrm{GKC} & \textrm{SAT} & \textrm{Avg.} \\\\
> > > \\hline
> > > \textrm{DELI} & \textbf{363.21} & \textbf{716.44} & \textbf{450.15} & \textbf{583.23} & \textbf{869.49} & \textbf{1181.75} & \textbf{483.42} & \textbf{663.96} \\\\
> > > \textrm{Retrieval CoT SC n=4} & 432.82 & 760.61 & 583.21 & 791.81 & 876.39 & 1377.06 & 810.68 & 804.65 \\\\
> > > \textrm{ReAct SC n=4} & 490.00 & 849.30 & 602.92 & 941.57 & 1161.56 & 1265.10 & 537.39 & 835.40 \\\\
> > > \end{array}
> > >
> > > As shown in the table, DELI requires less output token number than Self-Consistency methods (n=4), since DELI involves fewer turn numbers on average and the two stages of DELI can refer to each other to provide a more refined process. As a result, DELI achieves better performance compared to baseline methods (Retrieval CoT and ReAct) with Self-Consistency (n=4) with fewer average output token numbers and average turn numbers.
> > >
> > > **7. I still think the novelty of the proposed method is limited. This approach is more about trading the time spent on inference for the performance improvement. In addition, Whether it is worth it remains unproven, considering (6).**
> > >
> > > The key difference between our work and previous work (e.g., ReAct) is that it can largely improve the reasoning accuracy by gradually refining the reasoning procedure. The reason that we take this motivation is based on the performance level of existing LLMs. Till now, LLMs are still struggling with complex reasoning tasks (e.g., solving math problems), especially for single-pass generation for error accumulation. We believe multi-pass generation or interaction has become a promising direction to explore the more powerful abilities of LLMs [1-4]. That is the foundation of LLM-based planning or agents. We naturally follow such a multi-pass approach. Our method is motivated by the specific issues found in our fine-grained evaluation, including error accumulation and the difficulty of incorporating tool use in the natural reasoning process. Besides, our method incorporates the combination of tool manipulation and natural language reasoning in an iterative manner, which has rarely been taken into account in existing work.
> > >
> > > Considering the inference cost, we believe that it is unfair to compare our approach with single-pass reasoning methods (e.g., CoT). Instead, a fair comparison should be with typical multi-pass approaches, such as CoT and ReAct augmented with Self-Consistency. As shown in Point 6 and Table 4, our method achieves better performance than ReAct with Self-Consistency under similar or fewer inference costs in all evaluated datasets.
> > >
> > > We appreciate that the reviewer has provided a very insightful comment on this issue. However, we believe that the novelty of this work should be evaluated based on the entire contribution (datasets, evaluation protocols and methodology). In addition, we strongly suggest that the inference cost should be discussed or compared in models with similar approaches or capacities, but not solely on the model complexity (by analogy, LLMs are often more costly than small-sized PLMs, but we may still prefer a capable LLM instead of a small PLM).
> > >
> > > [1] Wang X, Wei J, Schuurmans D, et al. Self-consistency improves chain of thought reasoning in language models[J]. arXiv preprint arXiv:2203.11171, 2022.
> > >
> > > [2] Shinn N, Labash B, Gopinath A. Reflexion: an autonomous agent with dynamic memory and self-reflection[J]. arXiv preprint arXiv:2303.11366, 2023.
> > >
> > > [3] Guo Y, Liang Y, Wu C, et al. Learning to program with natural language[J]. arXiv preprint arXiv:2304.10464, 2023.
> > >
> > > [4] Zheng C, Liu Z, Xie E, et al. Progressive-hint prompting improves reasoning in large language models[J]. arXiv preprint arXiv:2304.09797, 2023.

---

> > > ### Author Response · Authors · 2023-08-30
> > > **Gentle Reminder for the Discussion**
> > >
> > > Dear Reviewer 73Fj,
> > >
> > > Thanks again for your careful reading and insightful feedback of our paper. We have tried our best to elaborate on the unclear points. We have discussed the significance of using mathematical reasoning datasets as a testbed combining tool manipulation and complex reasoning. We also have conducted an analysis on the issue of incorrect steps leading to correct answers, and found that the issue is quite uncommon in the evaluation. Besides, we have followed your suggestion to report the average output token number of methods and further show the effectiveness of DELI with fewer inference costs. In addition, we have further discussed the novelty of DELI compared to ReAct and multi-pass generation methods.
> > >
> > > Since the author-reviewer discussion period stage is coming to an end today, we are more than happy to hear your comments and address any of your further concerns during the remaining time.
> > >
> > > Best,
> > >
> > > Authors

---

> > > > ### Comment · Reviewer_73Fj · 2023-08-30
> > > >
> > > > Hi, thanks for your responses.
> > > >
> > > > (4) I'm not fully convinced by your argument that "though only the math domain is considered, we argue that the math reasoning performance of a LLM can actually reflect its comprehensive ability." If this is true, we only need to use the math ability to measure LLMs.
> > > >
> > > > (5) I do not understand why this is "a fundamental issue in all the reasoning-related datasets". To me, this is a fundamental limitation only in the created dataset in this work that requires the intermediate step annotation.
> > > >
> > > > (6) I do not understand why setting the turn number of self-consistency and ReAct to 4. These two approaches may directly solve this task in a single turn in their original implementation.  However, DELI requires multiple rounds of interactions, which is inefficient in some sense.

---

### Official Review · Reviewer_mfSt · 2023-07-18
**A new and useful dataset with a novel approach**

**Rating:** 7
**Confidence:** 3
**Correctness:** The claims made by the authors in the…
**Clarity:** This paper is well-written.

**Strengths:**

(1) This paper constructs a new dataset named CARP with formulated annotation of intermediate reasoning steps for systematically evaluating LLMs in solving computation-intensive math problems, and devise interfaces with practical functionalities to help LLMs.

(2) This paper proposes a novel approach that can deliberate and correct the reasoning steps of LLMs with tool interfaces. Such a way is similar to the solution checking process of humans and very interesting.

(3) The experiments are comprehensive and reasonable.

**Additional Feedback:**

(1) Why does DELI exhibit highest Fail@last and higher Fail@middle?  What is the mechanism behind failure? Do these failure cases reveal the limitations of the model?

(2) A math problem may have different solutions to solve it. How should you handle intermediate steps in the dataset construction, when there are multiple solutions for a math problem? If only one solution approach is retained in the dataset, will using different solution steps affect the "fail@where" metric? If multiple solution approaches are retained, how can we exhaust all possible approaches to a mathematical question?

**Documentation:**

Yes.

**Ethics:**

No.

**Limitations:**

The authors have adequately addressed the limitations.

**Opportunities For Improvement:**

Table 2 appears before Table 1 in the manuscript text. It is recommended to adjust the position of the tables.

**Relation To Prior Work:**

Yes.

**Summary And Contributions:**

This paper focuses on computation-intensive math reasoning. Most existing math reasoning datasets may not be able to fully evaluate and analyze the ability of LLMs in manipulating tools and performing reasoning. To address the issue, the authors construct CARP, a new Chinese dataset consisting of 4,886 computation-intensive algebra problems with formulated annotations on intermediate steps. Unfortunately, existing LLM with CoT prompting are all prone to make mistakes at the early steps in CRAP. Then, the authors propose a new approach (DELI) that can deliberate the reasoning steps with tool interfaces. Experimental results on CARP and six other datasets show that the proposed DELI mostly outperforms competitive baselines, and can further boost the performance of existing CoT methods.

---

> ### Author Response · Authors · 2023-08-21
> **Response to the Concerns of Reviewer**
>
> We sincerely thank the reviewer for the insightful suggestion and appreciate the positive feedback. We have listed our response to your concerns as follows. If you also have any other questions, please feel free to let us know. We will continue to try our best to answer for you.
>
> **1. Table 2 appears before Table 1 in the manuscript text. It is recommended to adjust the position of the tables.**
>
> Thank you for pointing out the issue. We will fix it in the updated version.
>
> **2. Why does DELI exhibit highest Fail@last and higher Fail@middle? What is the mechanism behind failure? Do these failure cases reveal the limitations of the model?**
>
> In this work, we propose two metrics for evaluating the intermediate reasoning process, including ExpAcc and Fail@where. ExpAcc is the major metric to evaluate the accuracy of intermediate results, which can measure how good a model is. While ***Fail@where denotes the rate of making the "first mistakes" in the first/middle/last step among all instances with incorrect answers***, which is just used to analyze the distribution of the first mistakes for the model. Thus, the sum of Fail@first, middle, and last is 1, and it is meaningless to directly compare the percentage values of them between models.
>
> In contrast, Fail@where is a very valuable metric to diagnose the errors of the LLM. In Section 2.2, we conduct the preliminary experiment on CARP, and reveal that LLMs are prone to make mistakes in the early steps (69% errors in the first step). Once LLMs make mistakes in the early steps, they rarely fix the errors in the following generation, which is one of the key limitations for LLMs to solve multi-step reasoning problems. To alleviate the issue, we enable LLMs to deliberate existing solutions from different perspectives (natural language reasoning and tool manipulation), which provides a mechanism for LLMs to fix their errors in early steps. As shown in Table 5, DELI can achieve significantly higher ExpAcc than baselines. Besides, DELI also achieves a relatively lower Fail@first but higher Fail@middle and Fail@last. It means that DELI postpones the "first mistakes" of the LLMs and fixes many errors in the early steps.
>
> **3. A math problem may have different solutions to solve it. How should you handle intermediate steps in the dataset construction, when there are multiple solutions for a math problem?**
>
> In fact, it is very hard to enumerate all the possible solutions for a problem, due to the diverse natural language expression and reasoning processes. In our work, we still try our best to give a fair comparison of all the possible solutions.
>
> First, we do not exactly match the words of the natural language, but mainly focus on the correctness of the derived math expressions. We hand-craft a number of postprocessing strategies to guarantee a comparison between math expressions as fair as possible, such as cleaning the expressions by string replacement and regular expressions, trying to recall more matching expressions with minor errors. For the answer comparison, following existing work[1], we first judge whether the answer strings exactly match. If not, we convert the given answers into SymPy objects and measure the math equivalence, to avoid the mismatching caused by formatting.
>
> Second, our proposed evaluation method for all the reasoning steps based on the CARP dataset, has considered the multiple reasonable solving processes. As all the possible reasonable paths just fork in the intermediate steps and merge in the later, we check the correctness of the reasoning steps from back to front.
>
> For example, to solve the problem ``Knowing that the fractional equation $-\\frac { 3 } { m + 6 } = \\frac { m } { x }$ with respect to $x$ has the same solution as the fractional equation $\frac { 3 } { 2 x } = \\frac { 1 } { x - 1 }$, then the value of $m ^ { 2 } - 2 m$ is ____ ?'', if the i-th reasoning step matched the annotated steps (e.g., $m=3$), its front steps, i.e., from the 1-th to (i-1)-th ones, are all regared as the corrrect ones, whatever they differ from the annotated ones (e.g., directly solving the solution $\frac { 3 } { 2 x } = \\frac { 1 } { x - 1 }$ to get $x=3$ or substituting $x=\frac{m(m+6)}{3}$ into $\frac { 3 } { 2 x } = \\frac { 1 } { x - 1 }$). It can fairly assign the same scores for multiple reasonable but different solutions.
>
> We are still seeking a good way to efficiently and accurately check the possible errors in the intermediate steps, both the natural language and the logic. But now, we do not have a good way to achieve this goal. A possible solution is to utilize the SOTA LLMs (e.g., GPT-4) to verify the solution step-by-step based on our annotation, but it is too expensive to be extensively utilized. Besides, due to the uncountable possible reasonable solutions, it is also hard to guarantee the correctness of the LLMs, even human beings. We leave this issue to our future work.

---

> ### Author Response · Authors · 2023-08-26
> **Gentle Reminder for the Discussion**
>
> Dear Reviewer mfSt,
>
> Thanks for your careful reading and insightful feedback of our paper. We have clarified the definition and the findings of the proposed metric Fail@where. Besides, we have added a discussion about how our evaluation method handles multiple possible reasoning paths. We would like to know whether you find our response satisfactory, or if there are more questions that we could clarify. Since the rebuttal stage is coming to an end, we are more than happy to hear your comments and address any of your further concerns during the remaining time.
>
> Best,
>
> Authors

---

### Official Review · Reviewer_snqK · 2023-07-19
**Computational dataset with intermediate steps annotations and novel method to improve the computational reasoning capability of LLMs**

**Rating:** 7
**Confidence:** 4
**Clarity:** The paper is well written.

**Strengths:**

- The authors have open-sourced a dataset, CARP, which is reasonably difficult and diverse unlike the existing elementary reasoning math datasets (e.g. GSM8K) but not too advanced (e.g. MATH, MMLU-STEM). This dataset comes with intermediate computation annotations that are useful for evaluating the intermediate steps, given that language models often make a mistake at an intermediate step before reaching to the final answer.

- The proposed approach, DELI, leads to non-trivial degree of improvement over various baselines on 7 computation-intensive math reasoning datasets.

- The authors have demonstrated that even the powerful closed-source LLMs still fail at basic computational problems, which makes their proposed dataset and method very convincing.

**Additional Feedback:**

N/A

**Correctness:**

The dataset creation, testing, and evaluation methodology are all sound. This paper appears correct about its claims.

**Documentation:**

The dataset is open-source and well-documented along with its pipeline.

**Ethics:**

Given the nature of this project, there seems no ethical concern with the submission.

**Limitations:**

The authors adequately addressed the limitations and potential negative societal impact of their work.

**Opportunities For Improvement:**

The major criticism I have with this work and other papers that focus on computational problems specifically is their lack of applicability to non-computational problems, though this may not be a valid criticism in the context of paper review for NeurIPS. While language models still struggle with computational problems, there are many non-computational math (e.g. Abstract Algebra) and non-math questions that require advanced reasoning capability. While the use of tools helps (e.g. calculator) even humans to solve these problems, humans can still solve these problems without tools, and it is not always guaranteed that there is a tool for a given problem. Hence, the important question is whether a model can solve these problems without task-specific human interventions.

**Relation To Prior Work:**

The paper clearly discusses how it relates to prior work.

**Summary And Contributions:**

The contributions can be summarized as follows:

- Constructing a new dataset named CARP with formulated annotation of intermediate reasoning steps for systematically evaluating LLMs in solving computation-intensive math problems, and devising interfaces with practical functionalities to help LLMs.

- Proposing DELI, a new approach that can deliberate and correct the reasoning steps of LLMs with tool interfaces.

- Conducting extensive experiments to show the superiority of our DELI over existing prompting 76 methods on 7 computation-intensive math reasoning datasets.

---

> ### Author Response · Authors · 2023-08-21
> **Response to the Concerns of Reviewer**
>
> We sincerely thank the reviewer for the insightful suggestion and appreciate the positive feedback. We have listed our response to your concerns as follows. If you also have any other questions, please feel free to let us know. We will continue to try our best to answer for you.
>
> **1. The major criticism I have with this work and other papers that focus on computational problems specifically is their lack of applicability to non-computational problems. While language models still struggle with computational problems, there are many non-computational math (e.g. Abstract Algebra) and non-math questions that require advanced reasoning capability. While the use of tools helps (e.g. calculator) even humans to solve these problems, humans can still solve these problems without tools.**
>
> Reasoning ability is one of the most high-level intelligence for machines. It is the key to opening the gate of the great future with automatic world knowledge discovery and decision-making with fewer human interventions. As the reviewer mentioned that there are many scenarios that require advanced reasoning ability, but the computation-intensive math reasoning task is one of the most complex ones, which needs to understand the abstract math theories and obscure concepts, and also well learn the rigorous deduction logic and accurate numerical computation. Therefore, the AI models for this task can also be useful references for other reasoning tasks.
>
> For the concern about using tools, in our work, the LLM is not mandatorily enforced to use the tool, but can freely choose to use the proper one or not. If the LLM can solve the problem by itself, it would invoke the "Thinking" interface, to just perform natural language generation. Note that in our approach, tools are mainly used to help extend the capabilities of LLMs, to handle the complicated computations where LLMs are prone to make mistakes. It is very effective in improving the performance of the individual LLM for many complex scenarios, including ours and other reasoning tasks, e.g., multi-hop knowledge reasoning. To verify it, we add an experiment on HotpotQA, a challenging QA dataset requiring models to find and reason over multiple supporting documents to answer. We implement the baseline ReAct and DELI on gpt-3.5-turbo (Aug.), and follow existing work [1] to provide two tool interfaces:
> > (1) Search[entity], which searches the exact entity on Wikipedia and returns the first paragraph if it exists. If not, it will return some similar entities to search.
> > (2) Lookup[keyword], which returns the next sentence containing keyword in the current passage.
>
> As shown in the table, DELI can also improve the performance of baseline methods in the multi-hop knowledge reasoning task, indicating its effectiveness in non-math tasks.
>
> \\begin{array} {|l|c|}
> \\hline
> \\textrm{Model} & \\textrm{EM} \\\\
> \\hline
> \\textrm{ReAct} & 23.80 \\\\ \\hline
> \\textrm{Random CoT} &  30.27  \\\\  \\hline
> \\textrm{DELI} & \\textbf{32.53}  \\\\
> \\end{array}
>
> [1] Yao S, Zhao J, Yu D, et al. React: Synergizing reasoning and acting in language models[J]. arXiv preprint arXiv:2210.03629, 2022.
>
> **2. It is not always guaranteed that there is a tool for a given problem. Hence, the important question is whether a model can solve these problems without task-specific human interventions.**
>
> Frankly speaking, it is not always guaranteed that there is a best-suited tool for a given problem. Whereas, tools should not merely be confined to the existing arsenal, as they can be expanded to encompass a vast array of resources and methodologies, such as databases, knowledge graphs, and crowd-sourced workers. All the external resources can be encapsulated into "new tools" with the help of human interventions. Therefore, as the reviewer mentioned, the greatest vision is that models possess the capability to perform general tasks without requiring specific human interventions. However, this vision has yet to be realized as even the most advanced LLMs (e.g., GPT-4) fall short of achieving this level of autonomy and effectiveness. In our future work, we are going to explore the way to enable LLMs to autonomously create their own tools that allow them to adapt to different scenarios.

---

> ### Author Response · Authors · 2023-08-26
> **Gentle Reminder for the Discussion**
>
> Dear Reviewer snqK,
>
> Thanks for your careful reading and insightful feedback of our paper. We have added a discussion of the concern of using tools. We also have conducted an experiment on the applicability of DELI to another reasoning scenario. Besides, we have added a discussion of the scalability of the tool manipulation. We would like to know whether you find our response satisfactory, or if there are more questions that we could clarify. Since the rebuttal stage is coming to an end, we are more than happy to hear your comments and address any of your further concerns during the remaining time.
>
> Best,
>
> Authors

---

### Official Review · Reviewer_mtC2 · 2023-07-20
**Interesting approach to analyze chain-of-thoughts, but some aspects could be explored further**

**Rating:** 7
**Confidence:** 3

**Strengths:**

With the rising interest in solving multi-steps reasoning problems using recursive calls to LLMs and formal tools, the paper proposes an interesting dataset to better analyze and decompose this process in the case of mathematical problems. The paper proposes a large number of empirical results. Beyond the dataset, the paper also introduces a new deliberation method, which outperforms comparable related work approaches not only on this dataset but also other math benchmarks. Finally, the DELI methods can be combined with existing approaches to further improve the results.

**Additional Feedback:**

N/A

**Clarity:**

Overall, the paper is well written and clear. Some elements are delegated to the appendix but may find a better place in the main section of the paper. Including  the related work section, which provides a good element of context regarding effort in the domain.

The authors give some metrics but do not explicitly say what is reported in Table 4.

**Correctness:**

It seems to me that the prompts mix English and Chinese. What is the reason behind it? Is there any impact of this choice?

It should be mentioned somewhere that comparing the generated answer with the reference is not trivial and requires some cleaning operations relying on manual rules, cleaning, and regular expressions in the code. Indeed, it seems one cause of error is that the LLM could simply output some incorrect latex code for the answer.

**Documentation:**

Yes, there is sufficient detail to support reproducibility.

**Ethics:**

No, I do not suspect any ethical concerns with the submission that warrant further discussion or review.

**Limitations:**

The authors rightly mention that they only tested one model, namely GPT-3.5 turbo. The model itself is not open-source and the dataset used for training is not publicly available. It would be interesting to have data points for other models in future work. I did some experiments with Baichen-7B, which resulted in far worse results, i.e. 17 acc on random COT. This could highlight the importance of the model size and pre-training conditions.

Line 137 mentions a “reasonable solving process should first deduce the intermediate equation [...]”. However, it seems that some
problems may have multiple (reasonable) possible solving processes?

**Opportunities For Improvement:**

There are still some limitations on the dataset use, i.e. not possible to analyze the type of reasoning errors. I re-run some of the paper experiments with smaller LLMs [2]. I observed that error often appends based on a bad interpretation of the problem. For example, instead of giving the condition under which $\sqrt{x + 2}$ is a valid expression, i.e. $x > -2$, the LLM outputs the condition under which the expression is not valid, i.e. $x \leq -2$. But this is not due to a “reasoning problem”, simply a bad reformulation of the problem itself. It could be good to use the annotations to better characterize the type of mistakes that are made.

The paper does not present any standard deviation for the results in Table 4. Given appendix D.3, temperature for decoding is set to 0 and top_p to 1. Does this mean that the model output does not contain any randomness for a given prompt? Yet, there could be some variations for the first line (random CoT) between two runs since examples are randomly sampled.

The paper does not discuss the impact of the prompt format and wording on the results. Paragraph line 242 suggests that some elements in the prompts were added to lessen inconsistency in intermediate results. However the impact is not discussed. Given recent related experiments (for example [1]), I expect the instruction formulation in the prompt may have some impact that could be interesting to discuss, especially in the light of the previous remark.

[1] https://huggingface.co/blog/evaluating-mmlu-leaderboard

[2] https://huggingface.co/baichuan-inc/Baichuan-7B

**Relation To Prior Work:**

Related work is very clear and well presented. Too bad it is in the appendix.

**Summary And Contributions:**

The paper introduces CARP, a dataset of annotated algebra problems. The dataset includes the solution of the problem together with intermediate steps required to obtain it. It is specifically designed to better study the ability of LLMs to perform step-by-step reasoning using chain-of-thought prompting methods. Observing the short-comings of current approaches, the paper introduces DELI, a new approach, which combines chain-of-thought prompting, similar examples retrieval, external tools manipulation, and iterative refinement of the solution. The paper shows the approach outperforms existing one on the CARP dataset as well as other mathematical datasets.

---

> ### Author Response · Authors · 2023-08-21
> **Response to the Concerns of Reviewer [Part 1]**
>
> We sincerely thank the reviewer for the insightful suggestion and appreciate the positive feedback. We have listed our response to your concerns as follows. If you also have any other questions, please feel free to let us know. We will continue to try our best to answer for you.
>
> **1. Concerns that the dataset is not possible to analyze the type of reasoning errors. It could be good to use the annotations to better characterize the type of mistakes that are made.**
>
> Frankly speaking, due to the uncountable error types in the machine-generated solution, it is hard to provide the best annotations that can analyze all the possible reasoning errors. In our CARP dataset, we mainly focus on the formulated annotations of the Expression Flow Graphs for the solutions, which is helpful to check and analyze the "errors of expressions and computation", two important issues for mathematical reasoning [1,2], especially for the computation-intensive task. Based on the annotations, we automatically analyze the types of the first incorrect step with fine-grained diagnosis on them, for the baseline and our approach.
>
> As shown in the following table, we can see several interesting findings. First, compared to Retrieval CoT,  DELI reduces the ratios of the errors of Equation/Inequation solving and computation (substitution and calculation). It indicates its effectiveness to correct the errors of expressions and computation within the solutions via iterative deliberations. Second, both methods are prone to make errors in natural language reasoning (including the reviewer-mentioned error). However, it requires specifically carefully designed annotations and programs for detailed analysis, which is not the main focus of this work. We would leave it into our future work, as the upgrade plan of our CARP dataset.
>
> \\begin{array} {|l|c|c|}
> \\hline
> \\textrm{Incorrect Step Type} & \\textrm{Retrieval CoT} & \\textrm{DELI}  \\\\
> \\hline
> \\textrm{Natural language reasoning} & 137 & 149  \\\\ \\hline
>         \\textrm{Equation/Inequation solving} & 79 & 28  \\\\ \\hline
>         \\textrm{Expression transformation} & 26 & 25  \\\\ \\hline
>         \\textrm{Equation/Inequation manipulation} & 24 & 21 \\\\ \\hline
>         \\textrm{Substitution} & 23 & 4  \\\\ \\hline
>         \\textrm{Calculation} & 3 & 1 \\\\ \\hline
>         \\textrm{Total} & 292 & 228 \\\\ \\hline
> \\end{array}
>
> [1] Qian J, Wang H, Li Z, et al. Limitations of language models in arithmetic and symbolic induction[J]. arXiv preprint arXiv:2208.05051, 2022.
>
> [2] Lu P, Qiu L, Yu W, et al. A survey of deep learning for mathematical reasoning[J]. arXiv preprint arXiv:2212.10535, 2022.
>
>
> **2. The paper does not present any standard deviation for the results in Table 4.**
>
> According to OpenAI API documentation [1], setting the temperature for decoding to 0 will make the outputs mostly deterministic, but a small amount of variability may remain. To test if the variation would affect the performance of approaches, we repeatedly run the experiment 3 times and report the average and standard deviation in the following table. As depicted in the table, most of the standard deviation values are less than 1.0, indicating less randomness of the results. Besides, our method consistently demonstrates superior performance compared to all baseline methods, showing its effectiveness.
>
> Note that we perform the extended experiments on gpt-3.5-turbo in August, whose performance is improved compared to gpt-3.5-turbo before the submission.
>
> \\begin{array} {|l|c|c|c|c|c|c|c|c|}
> \\hline
> \\textrm{Method} & \\textrm{CARP} & \\textrm{Algebra}   & \\textrm{Prealgebra} & \\textrm{CP}  & \\textrm{NT}  & \\textrm{GKC} & \\textrm{SAT} & \\textrm{Avg.} \\\\
> \\hline
> \\textrm{Retrieval CoT} & 69.74 (\\pm 0.33) & 60.94 (\\pm 0.18) & 59.09 (\\pm 0.24) & 33.76 (\\pm 0.00) & 30.12 (\\pm 0.65) & 13.56 (\\pm 0.85) & 73.33 (\\pm 0.95) & 48.65 (\\pm 0.46) \\\\ \\hline
> \\textrm{ReAct} & 63.28 (\\pm 0.56) & 56.33 (\\pm 0.21) & 55.68  (\\pm 0.64) & 38.47 (\\pm 0.53) & 32.34 (\\pm 0.77) & 18.36 (\\pm 0.98) & 61.67 (\\pm 1.46) & 46.59 (\\pm 0.74) \\\\ \\hline
> \\textrm{DELI} & \\textbf{75.34} (\\pm 0.16) & \\textbf{66.29} (\\pm 0.48) & \\textbf{62.21} (\\pm 0.52) & \\textbf{41.42} (\\pm 0.74) & \\textbf{36.22} (\\pm 0.45) & \\textbf{18.92} (\\pm 0.49) & \\textbf{75.75}  (\\pm 1.05) & \\textbf{53.74} (\\pm 0.56) \\\\ \\hline
> \\end{array}
>
> [1] https://platform.openai.com/docs/models/gpt-3-5

---

> ### Author Response · Authors · 2023-08-21
> **Response to the Concerns of Reviewer [Part 2]**
>
> **3. The paper does not discuss the impact of the prompt format and wording on the results. I expect the instruction formulation in the prompt may have some impact that could be interesting to discuss, especially in the light of the previous remark.**
>
> In our proposed DELI, we carefully design the iterative deliberation strategy to gradually refine the quality of generated solution, but do not spend too much time on the design of the prompt format. We mainly follow existing guidance [1,2] that adds the commonly-used ingredients into the prompt, and our past experiments find that it works well for all LLMs. Besides, our experiments also reveal that it is hard to design the best prompt for all LLMs, even an LLM with different versions. And the effectiveness of all components is also unstable across different LLMs.
>
> We show the results on gpt-3.5-turbo (0613) and (August) in the following table. First, we can see that removing the instruction or format restriction would lead to slight performance changes in our approach, and the change tendencies are unstable across the two versions of gpt-3.5-turbo. Although we try two very different variations of the prompts (i.e., with more detailed or simplified prompts), the value also changes slightly and unstably, making it hard to leave a precise conclusion about the impact. Thus, we think that our used prompt has reached a good point with the deliberation strategy and in-context exemplars, and it is unnecessary to spend much time on prompt engineering for the possible trivial but unstable improvement.
>
> \\begin{array} {|l|l|c|c|}
> \\hline
> \\textrm{Model} & \\textrm{Method} & \\textrm{CARP} & \\textrm{CP}  \\\\
> \\hline
> \\textrm{gpt-3.5-turbo-0613} & \\textrm{DELI} & 76.84 & 41.56  \\\\
> & \\textrm{\\quad w/o task instruction} &  77.05 & 41.98  \\\\
> & \\textrm{\\quad w/o format restriction} & 76.33 & 41.35  \\\\
> & \\textrm{\\quad w/ more detailed prompt} & 76.64 & 40.72   \\\\
> & \\textrm{\\quad w/ more simplified prompt} & 77.05 & 41.56 \\\\\\hline
> \\textrm{gpt-3.5-turbo (Aug.)} & \\textrm{DELI} & 75.34 & 41.42   \\\\
> & \\textrm{\\quad w/o task instruction} &  75.10 & 42.62  \\\\
> & \\textrm{\\quad w/o format restriction} & 75.82 & 41.77  \\\\
> & \\textrm{\\quad w/ more detailed prompt} & 76.33 & 40.30   \\\\
> & \\textrm{\\quad w/ more simplified prompt} & 76.43 & 41.56 \\\\\\hline
> \\end{array}
>
>
> [1] Yao S, Zhao J, Yu D, et al. ReAct: Synergizing Reasoning and Acting in Language Models[C]//The Eleventh International Conference on Learning Representations. 2023.
>
> [2] Wu C, Yin S, Qi W, et al. Visual chatgpt: Talking, drawing and editing with visual foundation models[J]. arXiv preprint arXiv:2303.04671, 2023.
>
> **4. The authors rightly mention that they only tested one model, namely GPT-3.5 turbo. The model itself is not open-source and the dataset used for training is not publicly available. It would be interesting to have data points for other models in future work**
>
> As existing work has shown that GPT-3.5-turbo has shown stronger mathematical reasoning performance among all LLMs, we conduct the experiment on it. Following the suggestion of the reviewer, we also implement baselines and our approach on other LLMs, including three closed-source LLMs (gpt-3.5-turbo (August), text-davinci-003, and claude-2 [1]) and an open-source LLM (Qwen-7B [2]). As shown in the following table, DELI also outperforms other baseline methods across various LLMs. Moreover, the best performance of DELI is achieved on gpt-3.5-turbo, indicating its strong reasoning capability.
>
> \\begin{array} {|l|l|c|c|}
> \\hline
> \\textrm{Model} & \\textrm{Method} & \\textrm{CARP} & \\textrm{CP}  \\\\
> \\hline
> \\textrm{gpt-3.5-turbo (Aug.)} & \\textrm{Retrieval CoT} & 69.74 & 33.76  \\\\
> & \\textrm{ReAct} &  63.28 & 38.4  \\\\
> & \\textrm{DELI} & \\textbf{75.34} & \\textbf{41.42} \\\\ \\hline
> \\textrm{claude-2} & \\textrm{Retrieval CoT} & 65.78 & 33.76  \\\\
> & \\textrm{ReAct} &  60.76 & 34.6  \\\\
>  & \\textrm{DELI} & \\textbf{74.49} & \\textbf{40.93} \\\\ \\hline
> \\textrm{text-davinci-003} & \\textrm{Retrieval CoT} & 53.48 & 25.53  \\\\
> & \\textrm{ReAct} &  54.92 & 22.78  \\\\
>  & \\textrm{DELI} & \\textbf{60.45} & \\textbf{29.32} \\\\ \\hline
> \\textrm{Qwen-7B} & \\textrm{Retrieval CoT} & 56.86 & 14.77  \\\\
> & \\textrm{ReAct} &  43.95 & 13.29  \\\\
>  & \\textrm{DELI} & \\textbf{58.4} & \\textbf{15.44} \\\\ \\hline
> \\end{array}
>
> [1] https://www.anthropic.com/index/claude-2
>
> [2] https://github.com/QwenLM/Qwen-7B

---

> ### Author Response · Authors · 2023-08-21
> **Response to the Concerns of Reviewer [Part 3]**
>
> **5. Line 137 mentions a “reasonable solving process should first deduce the intermediate equation [...]”. However, it seems that some problems may have multiple (reasonable) possible solving processes?**
>
> We are sorry for the confusion written in our paper. Limited by the annotation cost, we only explicitly annotate one of the solving processes of the problem. But our proposed evaluation method for all the reasoning steps based on the CARP dataset, has considered the multiple reasonable solving processes. As all the possible reasonable paths just fork in the intermediate steps and merge later, we check the correctness of the reasoning steps from back to front.
>
> For example, given the problem ``Knowing that the fractional equation $-\\frac { 3 } { m + 6 } = \\frac { m } { x }$ with respect to $x$ has the same solution as the fractional equation $\frac { 3 } { 2 x } = \\frac { 1 } { x - 1 }$, then the value of $m ^ { 2 } - 2 m$ is ____ ?'', if the i-th reasoning step matched the annotated steps (e.g., $m=3$), its front steps, i.e., from the 1-th to (i-1)-th ones, are all regared as the corrrect ones, whatever they differ from the annotated ones (e.g., directly solving the solution $\frac { 3 } { 2 x } = \\frac { 1 } { x - 1 }$ to get $x=3$ or substituting $x=\frac{m(m+6)}{3}$ into $\frac { 3 } { 2 x } = \\frac { 1 } { x - 1 }$). It can fairly assign the same scores for multiple reasonable but different solutions.
>
> **6. It seems to me that the prompts mix English and Chinese. What is the reason behind it? Is there any impact of this choice?**
>
> In this work, as our experiments are conducted on both Chinese and English datasets, we have to use the corresponding language for the majority of the prompts. But we keep the system prompt and the prefix of tool invocations (such as `Action` and `Output`) in English, as our past experiments have shown that their Chinese versions are not more effective than our mixed ones. We report the experimental results of the baseline and DELI with the prompts all in Chinese in the following table. We can see that using the Chinese system prompt and prefix may hurt the performance. A possible reason is that GPT-3.5-turbo is trained mostly with English data, which endows it to well follow English instructions.
>
> \\begin{array} {|l|l|c|}
> \\hline
> \\textrm{Model} & \\textrm{System Prompt/Prefix} & \\textrm{CARP}  \\\\
> \\hline
> \\textrm{Retrieval CoT} & \\textrm{English} & \\textbf{69.74}  \\\\
> & \\textrm{Chinese} &  68.44  \\\\ \\hline
> \\textrm{DELI} & \\textrm{English} & \\textbf{75.34}  \\\\
> & \\textrm{Chinese} &  73.16  \\\\
> \\end{array}
>
> **7. Comparing the generated answer with the reference is not trivial and requires some cleaning operations relying on manual rules, cleaning, and regular expressions in the code. Indeed, it seems one cause of the error is that the LLM could simply output some incorrect latex code for the answer.**
>
> In fact, as LLMs are prone to make minor mistakes that cause incompatible errors with the interfaces, we indeed prepare several hand-crafted post-processing rules, such as cleaning the expressions by string replacement and regular expressions, trying to recall more matching expressions with minor errors. In practice, such a way can save most of the minor mistakes as the reviewer mentioned.
>
> For the answer comparison, following existing work [1], we first judge whether the answer strings exactly match. If not, we convert the given answers into SymPy objects and measure the math equivalence, to avoid the mismatching caused by formatting (e.g., strings and numbers).
>
> [1] Lightman H, Kosaraju V, Burda Y, et al. Let's Verify Step by Step[J]. arXiv preprint arXiv:2305.20050, 2023.
>
> **8. Related work is very clear and well presented. Too bad it is in the appendix.**
>
> Thank the reviewer for pointing out this issue. We will add the related work section to the main paper.
>
> **9. The authors give some metrics but do not explicitly say what is reported in Table 4.**
>
> Thank you for pointing out this issue. The metric reported in Table 4 is the Accuracy of the final answer. As described in the response to point 7, we also follow existing work to address the possible format mismatching issue when computing the Accuracy of the final answer.

---

> ### Author Response · Authors · 2023-08-26
> **Gentle Reminder for the Discussion**
>
> Dear Reviewer mtC2,
>
> Thanks for your careful reading and insightful feedback of our paper. We have added automatic analysis of the types of incorrect steps. We have also presented the standard deviation of baselins and our method. Besides, we have conducted experiments on the impact of wording and formulation of prompts and the impact of Chinese and English prompts. We also have added experiments on the generalization of DELI for other closed-source and open-source LLMs. Moreover, we have clarified the consideration for evaluating different reasoning paths in our proposed evaluation method. We have also clarified the process of the answer comparison. We have tried our best to elaborate on the unclear points. We would like to know whether you find our response satisfactory, or if there are more questions that we could clarify. Since the rebuttal stage is coming to an end, we are more than happy to hear your comments and address any of your further concerns during the remaining time.
>
> Best,
>
> Authors

---

> > ### Comment · Reviewer_mtC2 · 2023-08-28
> >
> > Thanks for taking the time for answering my concerns and running additional experiments, in particular the standard deviations in Table 4 and the comparison with open-source LLMs. I raised my rating from 6 to 7 accordingly.

---

> > > ### Author Response · Authors · 2023-08-28
> > > **Appreciate your new response and updated score**
> > >
> > > Dear Reviewer mtC2,
> > >
> > > **We sincerely thank you for the positive reply and the updated score!**
> > >
> > > Thanks for your constructive review. Your review really helped us greatly in improving our paper, and we are truly grateful for your comments. We are very happy to see that you raised the score for our response. And we sincerely appreciate your time and effort in reviewing our paper and reading our comments.
> > >
> > > Best,
> > >
> > > Authors

---

### Official Review · Reviewer_fi3X · 2023-07-31

**Rating:** 7
**Confidence:** 4
**Correctness:** Yes
**Clarity:** Yes

**Strengths:**

- An interesting new CoT dataset for Chinese
- Inspiring findings on how LLMs fail in CoT reasoning
- DELI, the newly proposed method to improve CoT reasoning, achieves strong performance and is novel as far as I know.

**Additional Feedback:**

I had a hard time following what “Deliberation” means, which seems strange wording.

**Documentation:**

Documentation is clear.

**Limitations:**

DELI might not work in zero-shot settings.

**Opportunities For Improvement:**

- Some of the wording might be misleading. From Section 3.2, the “Deliberation with Tool Manipulation” paragraph seems more like a new prompting format instead of actually using the tools. Could the authors clarify? If DELI is not actually using any external tool (e.g., a calculator or Python interpreter), please change the “tool manipulation” framing.
- A lot of stuff is going on in this one, which is good. But the two main parts (CARP and DELI) seem a bit detached. More specifically, CARP identifies the early-failure issues, but it is not intuitive to me why DELI would be a remedy for this (despite the evidence in Table 5). The paper can do a better job connecting these two parts.
- I would appreciate an honest discussion on the additional inference overhead coming with DELI. Since DELI requires multiple inference rounds, would self-consistency be a reasonable baseline to compare to?
- A lot of experimental details are missing. Most importantly, what is the LLM used in Section 4? Does DELI generalize to other LLMs?
- The paper provides little detail about the annotation process.

**Relation To Prior Work:**

Discussion of prior work and compared baselines is complete to the best of my knowledge.

**Summary And Contributions:**

This paper builds a new Chines dataset focusing on CoT multistep reasoning for algebra problems (CARP). Four LLMs are tested out on the newly-proposed dataset, showing that they are prone to mistakes in the early stages of the reasoning chains. Answering this issue, the paper further proposes to augment the LLMs with retrieval components that iteratively improve the CoT reasoning accuracy through retrieved examplars. Experiments show that the proposed method outperforms strong baselines.

---

> ### Author Response · Authors · 2023-08-21
> **Response to the Concerns of Reviewer [Part 1]**
>
> We sincerely thank the reviewer for the insightful suggestion and appreciate the positive feedback. We have listed our response to your concerns as follows. If you also have any other questions, please feel free to let us know. We will continue to try our best to answer for you.
>
> **1. Some of the wording might be misleading. From Section 3.2, the “Deliberation with Tool Manipulation” paragraph seems more like a new prompting format instead of actually using the tools. Could the authors clarify? If DELI is not actually using any external tool (e.g., a calculator or Python interpreter), please change the “tool manipulation” framing**
>
> We will revise the title of the paragraph to "Deliberation with Tool Manipulation Prompting". Actually, our DELI relies on our designed interfaces (in Table 6 of the supplemental material) to indirectly manipulate tools, where we design proper prompts to guide the LLM for generating the arguments of the invoked interface to manipulate external tools. As the prompting strategies are the key to our approach, we also revise the title of the next paragraph as "Deliberation with Chain of Thought Prompting".
>
> **2. A lot of stuff is going on in this one, which is good. But the two main parts (CARP and DELI) seem a bit detached. More specifically, CARP identifies the early-failure issues, but it is not intuitive to me why DELI would be a remedy for this (despite the evidence in Table 5). The paper can do a better job connecting these two parts.**
>
> Our work focuses on the tool-augmented computation-intensive task, which has not been fully studied in existing work. Thus, we aim to provide useful data resources and practical suggestions for improving the development of this direction, resulting in a high-quality dataset CARP and the method DELI. In CARP, we provide formulated annotations based on Expression Flow Graphs, to support the evaluation of the intermediate computation steps. Through the preliminary experiment on CARP, we see that LLMs with the CoT method are prone to make mistakes in the early steps (69% errors in the first step). Based on this finding, we list the possible reasons why the CoT prompting method on LLMs perform not well on computation-intensive task:
> > (1) CoT method can not revise the incorrect results of LLMs after generating it, and the errors will be accumulated, resulting in incorrect final answers;
> > (2) it is hard to directly integrate CoT method with external tools, as invoking tools would interrupt the generation process of LLMs;
>
> To address them, we propose DELI, a prompting method that can deliberate and refine the generated solution. We also list its advantages to handle the above issues:
> > (1) DELI can iteratively check and refine the generated result, which makes it able to correct the mistakes of LLMs and avoids error accumulation;
> > (2) DELI iterates the deliberation processes using CoT prompting method and tool manipulation prompting, respectively, which can benefit from the two prompting methods without conflict of interruption.
>
> Therefore, our DELI can gradually improve the quality of the solution via the two-view iterative deliberation, outperforming CoT and other competitive prompting methods. Following your suggestion, we will revise the paper to better represent the relations between CARP and DELI.

---

> ### Author Response · Authors · 2023-08-21
> **Response to the Concerns of Reviewer [Part 2]**
>
> **3. An honest discussion on the additional inference overhead coming with DELI, and its comparison with self-consistency method**
>
> In DELI, the additional inference overhead mainly derives from the iterations of the two deliberation processes. To reduce the overhead, in addition to the maximum turns, we also design the stop condition that once the result before and after the deliberation iteration is not changed (indicating it has converged), we stop the iterations. We have compared existing iterative baseline methods (such as Learning to Program, PHP, and Iterative CoT/ReAct) in Table 4. In practice, we find that our DELI only requires less than 4 iterations on average. We report the average turn numbers of all evaluated datasets as follows:
>
> \begin{array} {|l|c|c|c|c|c|c|c|}
> \hline
> \textrm{Method} & \textrm{CARP} & \textrm{Algebra}   & \textrm{Prealgebra} & \textrm{CP}  & \textrm{NT}  & \textrm{GKC} & \textrm{SAT} \\\\
> \\hline
> \textrm{DELI} & 3.52 & 3.49 & 3.27 & 3.29 & 3.32 & 3.72 & 3.97 \\
> \end{array}
>
> Besides, following the suggestion of the review, we also add experiments based on gpt-3.5-turbo (Aug.) to compare our method with Self-Consistency of Retrieval CoT and ReAct in the following table, where we follow PAL that sets the temperature as 0.7, and set the number of predicted solutions as 4 for a fair comparison under the same inference overhead.
>
> \\begin{array} {|l|c|c|c|c|c|c|c|}
> \\hline
> \\textrm{Method} & \\textrm{CARP} & \\textrm{Algebra}   & \\textrm{Prealgebra} & \\textrm{CP}  & \\textrm{NT}  & \\textrm{GKC} & \\textrm{SAT} & \\textrm{Avg}. \\\\
> \\hline
> \\textrm{Retrieval CoT SC n=4} & 70.49 & 65.37 & \\textbf{62.69} & 37.34 & \\textbf{36.48} & 11.86 & 75.00 & 51.32 \\\\ \\hline
> \\textrm{ReAct SC n=4} & 64.56 & 56.11 & 56.83 & 38.4 & 30.74 & 17.80 & 65.00 & 47.06 \\\\ \\hline
> \\textrm{DELI} & \\textbf{75.34} & \\textbf{66.29} & 62.21 & \\textbf{41.42} & 36.22 & \\textbf{18.92} & \\textbf{75.75} & \\textbf{53.74} \\\\ \\hline
> \\end{array}
>
> As shown in the table, our DELI outperforms the methods using self-consistency on most datasets. The reason is that the computation-intensive task is very difficult and the generated reasoning paths of LLMs are hard to consistently hit the correct answer, especially since they are easy to make mistakes in the early steps leading to diverse wrong answers. As a comparison, our DELI can deliberate and refine the generated solution from two different perspectives, which is more suitable to handle the error-prone computation-intensive task.
>
> Note that Self-Consistency and DELI are mutually orthogonal methods, which focus on widening and deepening the exploration of the correct answer. Therefore, it is promising to combine both methods to achieve better performance. We evaluate the Self-Consistency of DELI on CARP and Number Theory. As shown in the table, Self-Consistency of DELI can further boost the performance of DELI.
>
> \\begin{array} {|l|c|c|}
> \\hline
> \\textrm{Method} & \\textrm{CARP} & \\textrm{NT}  \\\\
> \\hline
> \\textrm{DELI} & 75.34 & 36.22  \\\\ \\hline
> \\textrm{DELI n=4} & \\textbf{77.66} & \\textbf{38.25} \\\\ \\hline
> \\end{array}

---

> ### Author Response · Authors · 2023-08-21
> **Response to the Concerns of Reviewer [Part 3]**
>
> **4. A lot of experimental details are missing. Most importantly, what is the LLM used in Section 4? Does DELI generalize to other LLMs**
>
> Due to the page limitation, we have to remove the experimental details in the appendix (page 17-18), where we present the detailed designs of the interfaces and their descriptions, the details of evaluating datasets, baselines and implementation of our approach. The LLM used in Section 4 and the whole paper is gpt-3.5-turbo in May.
>
> To test the generality of our method, we implement it on three popular closed-source LLMs (i.e., gpt-3.5-turbo in August, claude-2 [1], text-davinci-003), and conduct experiments on CARP and Count. & Prob. datasets. In addition to them, we also select an open-source LLM with fewer parameters, Qwen-7B [2], as it has shown strong math reasoning performance among LLMs with the same scale. We report the results in the following table, where we can see that DELI also outperforms the two baselines across various LLMs, indicating the effectiveness and generality of our approach.
>
> \\begin{array} {|l|l|c|c|}
> \\hline
> \\textrm{Model} & \\textrm{Method} & \\textrm{CARP} & \\textrm{CP}  \\\\
> \\hline
> \\textrm{gpt-3.5-turbo (Aug.)} & \\textrm{Retrieval CoT} & 69.74 & 33.76  \\\\
> & \\textrm{ReAct} &  63.28 & 38.4  \\\\
> & \\textrm{DELI} & \\textbf{75.34} & \\textbf{41.42} \\\\ \\hline
> \\textrm{claude-2} & \\textrm{Retrieval CoT} & 65.78 & 33.76  \\\\
> & \\textrm{ReAct} &  60.76 & 34.6  \\\\
>  & \\textrm{DELI} & \\textbf{74.49} & \\textbf{40.93} \\\\ \\hline
> \\textrm{text-davinci-003} & \\textrm{Retrieval CoT} & 53.48 & 25.53  \\\\
> & \\textrm{ReAct} &  54.92 & 22.78  \\\\
>  & \\textrm{DELI} & \\textbf{60.45} & \\textbf{29.32} \\\\ \\hline
> \\textrm{Qwen-7B} & \\textrm{Retrieval CoT} & 56.86 & 14.77  \\\\
> & \\textrm{ReAct} &  43.95 & 13.29  \\\\
>  & \\textrm{DELI} & \\textbf{58.4} & \\textbf{15.44} \\\\ \\hline
> \\end{array}
>
> [1] https://www.anthropic.com/index/claude-2
>
> [2] https://github.com/QwenLM/Qwen-7B
>
>
> **5. The paper provides little detail about the annotation process.**
>
> We have shown the whole process of the annotation in Section 2.1. But due to the page limitation, we have to remove the details in the appendix (page 21-26), where we show the details about motivation, composition, collection process, preprocessing/cleaning/labeling, uses, distribution, and maintenance. Besides, we also show the annotation platform provided to the annotators to inspire researchers to re-implement our annotation process.
>
> **6. DELI might not work in zero-shot settings.**
>
> To validate the effectiveness of DELI in the zero-shot setting, we add experiments to test our method based on gpt-3.5-turbo (August) in the zero-shot setting on CARP and Count. & Prob datasets. The prompt in the zero-shot setting is modified from the few-shot prompt. We remove exemplars in the few-shot prompt and provide a more detailed instruction and format description.
>
> \\begin{array} {|l|l|c|c|}
> \\hline
> \\textrm{Model} & \\textrm{Method} & \\textrm{CARP} & \\textrm{CP}  \\\\
> \\hline
> \\textrm{gpt-3.5-turbo (Aug.)} & \\textrm{Zero-Shot Retrieval CoT} & 56.25 & 18.35  \\\\
> & \\textrm{Zero-Shot ReAct} &  40.16 & 25.74  \\\\
> & \\textrm{Zero-Shot DELI} & \\textbf{67.11} & \\textbf{29.11} \\\\ \\hline
> \\end{array}
>
> As shown in the result, DELI can also improve the performance of CoT in the zero-shot setting. Whereas, the performance of all approaches is not better than the few-shot settings. It indicates that few-shot exemplars are more helpful to guide LLMs for following instructions, than human-crafted zero-shot prompts. Besides, as zero-shot prompts mainly rely on the hand-crafted description and instruction, they are very sensitive to the LLMs and may not work across different LLMs, even fail after a minor update of the LLM. Thus, in practice, we suggest utilizing the few-shot setting for this task.
>
> **7. I had a hard time following what “Deliberation” means, which seems strange wording.**
>
> In this work, we use the word "Deliberation" to reflect that the LLM can self-check and refine its generated solution repeatedly, until the result converges. Such a process is like a human who carefully thinks and deliberates the possible mistakes in her or his written solutions. To help better understand its meaning, we will add footnotes and explanations in our paper to give a better explanation.

---

> ### Author Response · Authors · 2023-08-26
> **Gentle Reminder for the Discussion**
>
> Dear Reviewer fi3X,
>
> Thanks for your careful reading and insightful feedback of our paper. We have tried our best to elaborate on the unclear points. We have clarified the usage of external tools, the connection between the proposed dataset and method, the details of the annotation process, and the wording of deliberation in this work. Besides, we have added experiments about the comparison between DELI and Self-Consistency, and the generalization of DELI to other LLMs. We would like to know whether you find our response satisfactory, or if there are more questions that we could clarify. Since the rebuttal stage is coming to an end, we are more than happy to hear your comments and address any of your further concerns during the remaining time.
>
> Best,
>
> Authors

---

### Author Response · Authors · 2023-08-28
**General Response and Gentle Reminder for the Discussion**

We sincerely thank the five reviewers for their insightful and constructive feedback. We have crafted individual responses for each reviewer.

We outline the key points addressed in our replies for the proposed dataset CARP:
1. To address the challenge of evaluating intermediate processes in various reasoning paths, our evaluation method involves error tracing from back to front, treating ancestors of a recalled expression as recalled in the reference graph. Please refer to our responses addressed to Reviewer fi3x, Reviewer mfSt, and Reviewer 73Fj.
2. For the connection between the proposed dataset and method, we observe that LLMs are prone to make mistakes in the early steps. We propose DELI which can deliberate and refine the generated solution from two perspectives to correct the mistakes of LLMs and avoid the error accumulation. Please refer to our response addressed to Reviewer fi3x.
3. For analyzing error types at a fine-grained level, we automatically analyze the types of the first incorrect step based on the annotation. Please refer to our response addressed to Reviewer mtC2.
4. For the motivation of our dataset, we aim to provide a high-quality dataset that can enrich the data resource in the task, and also provide formulated annotations about the intermediate computation steps, which is useful to study the mistakes of LLMs in the intermediate steps. Please refer to our response addressed to Reviewer 73Fj.

We outline the key points addressed in our replies for the proposed method DELI
1. We add experiments to evaluate DELI based on multiple LLMs, which show that DELI improves baseline methods based on various LLM backbones. Please refer to our responses addressed to Reviewer fi3x and Reviewer mtC2.
2. We conduct an experiment to test DELI in the zero-shot setting, which shows that DELI consistently outperforms baselines in both zero/few-shot settings. Please refer to our response addressed to Reviewer fi3x.
3. We add experiments to compare DELI with Self-Consistency of two baseline methods, which shows that DELI outperforms Self-Consistency on most datasets. Please refer to our responses addressed to Reviewer fi3x and Reviewer 73Fj.
4. We add standard variation to further show the effectiveness of DELI. Please refer to our response addressed to Reviewer mtC2.
5. We add a variant study of the format and the wording of prompts, which shows that DELI is stable for different prompts. Please refer to our response addressed to Reviewer mtC2.
6. We add an experiment to evaluate DELI on the multi-hop knowledge reasoning task, which shows that DELI can also improve CoT and ReAct in other complex reasoning scenarios. Please refer to our response addressed to Reviewer snqK and Reviewer 73Fj.
7. For the novelty of DELI compared to ReAct, the major difference is that DELI incorporates the deliberation strategy to rethink and refine the generated solutions from two perspectives, while ReAct adopts the one-pass generation. Please refer to our response addressed to Reviewer 73Fj.

For more details, please refer to the separate responses for reviewers.

As the rebuttal phase is nearing its conclusion, we remain open to your feedback and are eager to address any additional concerns you may have within the remaining timeframe. Your comments and input are greatly appreciated.

---

### Author Response · Authors · 2023-08-30
**Grateful Acknowledgment to Reviewers**

We extend our genuine appreciation to the five reviewers for dedicating their time to offer feedback and suggestions on our work. Your comments have provided valuable insights that will aid in enhancing our paper. Since the author-reviewer discussion period is drawing to a close, we regret that we might not have access to address any new points after the deadline. We look forward to the possibility of engaging in future discussions.

---

### Decision · Program_Chairs · 2023-09-22

**Decision:**

Accept (Poster)

**Comment:**

This paper offers an interesting dataset for evaluation in a challenging domain (algebra problems) and will likely be helpful for those researching LLMs and tool use.  I also agree with the AC that "while not emphasized in the paper, I think this linguistic pluralism in this area is quite valuable."